

# Degradation of net primary production in a semi-arid rangeland

**H. Jackson[1] and S. D. Prince[1]**

[1]University of Maryland, College Park, Maryland, 20742, U.S.A.

*Correspondence to*: Hasan Jackson (hjackso1@umd.edu)

**Abstract.** Anthropogenic land degradation affects many biogeophysical processes
including reductions of net primary production (NPP). Degradation occurs at scales from
small fields to continental and global. While measurement and monitoring of NPP in
small areas is routine in some studies, for scales larger than 1km$^2$, and certainly global,
there is no regular monitoring and certainly no attempt to measure degradation.
Quantitative and repeatable techniques to assess the extent of deleterious effects and
monitor changes are needed to evaluate its effects on, for example, economic yields of
primary products such as crops, lumber and forage, and as a measure of land surface
properties which are currently missing from dynamic global vegetation models,
assessments of carbon sequestration and land surface models of heat, water, and carbon
exchanges. This study employed the Local NPP Scaling (LNS) approach to identify
patterns of anthropogenic degradation of NPP in the Burdekin Dry Tropics (BDT) region
of Queensland, Australia from 2000 to 2013. The method starts with land classification
based on the environmental factors presumed to control (NPP) to group pixels having
similar potential NPP. Then, satellite remotely sensing data were used to compare actual
NPP with its potential. The difference in units of mass of carbon and percentage loss was
the measure of degradation. The method is limited spatially only by the capacity to
classify the land. The entire BDT (7.45x10$^6$ km$^2$) was investigated at a spatial resolution
of 250x250m. The average annual reduction in NPP due to anthropogenic land
degradation in the entire BDT was -2.14 MgC m$^{-2}$ year$^{-1}$ or 17% of the non-degraded
potential, and the total reduction was -214 MgC year$^{-1}$. Extreme average annual losses of
524.8 gC m$^{-2}$ year$^{-1}$ were detected. Approximately 20% of the BDT was classified as
'degraded'. Varying severities and rates of degradation were found among the river
basins, of which the Belyando and Suttor were highest. Inter-annual, negative trends in
reductions of NPP, occurred in 7% of the entire region, indicating on-going degradation.
There was evidence of areas that were in a permanently degraded condition. The findings
provide strong evidence and quantitative data for reductions in NPP related to
anthropogenic land degradation in the BDT.





## 1 Introduction

Land degradation is a deleterious process in which unfavorable conditions for humans occur (Pickup, 1998, 1996; Safriel, 2007; Safriel and Adeel, 2005) as a result of direct and indirect human and natural processes. In drylands (aridity index < 0.65), poor land management such as excessive cultivation, overgrazing and unmanaged fires have far reaching effects on biogeophysical processes (Prince, 2002). While degradation is always undesirable, there is evidence that, in some cases, it cannot be reversed (Prince, 2016) when the causes are removed – a much more serious outcome. However, it is not known how widespread this condition is. There are many other aspects of dryland degradation that are little understood, including its location, severity and actions needed for remediation (Reynolds et al., 2007) or, at least, to prevent a net increase (Lal et al., 2012; UNCCD, 2012). The extent of soil or pasture degradation through overgrazing, anywhere in the world, has been estimated by experts' subjective opinion, rather than systematic quantitative criteria (Gifford, 2010).

'Degradation' implies an undesirable condition compared with a starting point (Prince 2016) but degraded compared to what? To detect a relative condition, a reference is needed, in this case not degraded (Bastin et al., 2012; Boer and Smith, 2003; Prince et al., 2007; Stoms and Hargrove, 2000) without which states of degradation have no meaning. However, the detection of non-degraded reference sites that are at their potential is problematic (Wessels et al., 2007). There are several approaches that seem reasonable but have severe limitations, particularly when applied to large areas: visual assessment of satellite imagery is entirely subjective and therefore unrepeatable; field surveys, such as the National Resources Inventory (Nusser and Goebel, 1997), are limited to small areas (Budde et al., 2004; O'Connor et al., 2001; Prince, 2004) that can be assessed by an evaluator on the ground; process modeling of potential production followed by comparison with actual production (Bai et al., 2008; Boer and Puigdefabregas, 2005) suffers from the need for data and parameters that are generally not available (Prince, 2002).

The particular type of degradation investigated here is anthropogenic reduction of net primary production (NPP) which, in addition to its own importance, is an indicator of a wider range of degradative processes (Prince, 2002) such as soil compaction, salinization, water and wind erosion that generally also reduce NPP (Pickup, 1996; Walker and Janssen, 2002). The objective of this study was to identify and characterize the extent and severity of degradation of vegetation productivity in the extensive rangelands, in excess of 10,000 km$^2$, of the Burdekin Dry Tropics (BDT) in Queensland, Australia. The Local NPP Scaling (LNS) method (Prince, 2004; Prince et al., 2009; Wessels et al., 2008) was used to address the problem of identification of reference sites. LNS starts with classification of the region into land capability classes (LCCs) in which the biogeophysical environment is, as near as possible, the same, so assessments are





made with areas of the same type and potential. The reference NPP is identified as the maximum value in each LCC, then the comparisons are made with this standard. Inaccuracies and even invalidity of the LNS technique can arise under certain conditions, although some methods are available that can minimize these, but they can never be

entirely prevented. On the other hand, bearing in mind the fundamental requirement for non-degraded comparison, and also that there is currently no other method available, LNS was used.

Specifically this study: (1) identified the spatial extent of non-degraded and degraded land; (2) distinguished significant land trends in inter-annual reductions in NPP;

and (3) linked total NPP reductions to specific land processes and states in the BDT.

## 2 Material and methods

### 2.1 Study area

The BDT region is located in north Queensland, Australia and covers approximately $7.45 \times 10^6$ km$^2$. The terrain is largely flat with gradually increasing

elevation inland (Mellick and Hanlon, 2005). Six large river basins are contained in the BDT (Figure 1): the Upper Burdekin ($2.26 \times 10^6$ km$^2$), Belyando ($2.08 \times 10^6$ km$^2$), Cape Campaspe ($1.18 \times 10^6$ km$^2$), Suttor ($1.07 \times 10^6$ km$^2$), Bowen Broken Bogie ($0.63 \times 10^6$ km$^2$) and Lower Burdekin ($0.23 \times 10^6$ km$^2$). Average seasonal rainfall varies spatially from 400 to 1500mm with a steep decreasing gradient from the coast inland. More than 70% of

rainfall falls during summer months (December-February) and runoff variability is high (Petheram et al., 2008; Rustomji et al., 2009). Discharge from rivers and creeks occurs in large pulses associated with intense but brief storms. During the study from 2000 to 2013, years with low (e.g. 2002-2007; $\leq$ 500mm year$^{-1}$) and high (e.g. 2008-2012; $\geq$ 600mm year$^{-1}$) accumulations occurred (Figure 2).

In the BDT, NPP is strongly influenced by regional variations in moisture availability (Hutley et al., 2000), fire frequency (Beringer et al., 2007), and soil properties. Native vegetation varies from dense to sparse forest to shrub-land and open grassland. Approximately 83% of the BDT is savanna consisting of mixed grass and trees. There are smaller areas that consist exclusively of shrubs (1%), grasses (8%), or

rain-fed crops (8%). The ratio of tree-to-grass cover is a defining attribute that differentiates local environments in savanna ecosystems (Accatino et al., 2010). The croplands, both irrigated and rain-fed, are found in northeastern, higher rainfall areas.

The major land use (85-90% of the BDT) is livestock production on unimproved pastures (Mellick and Hanlon, 2005). According to the State of Queensland (2011),

approximately 12% of the BDT has grazing practices likely to result in degradation.





### 2.2 Land capability classification

Land capability classes (LCCs) are areas that are homogeneous with respect to the selected environmental factors. The factors used here were meteorological, soil, and vegetation. The Australian Bureau of Meteorology distributes daily, synoptic weather
reports consisting of rainfall (Weymouth et al., 1999), minimum and maximum temperature, water vapor pressure deficit at 9am and 3pm, and solar exposure (Jones et al., 2009), gridded at 5x5km spatial resolution. Daily inputs were summed for the growing season from November to April and rescaled to 250x250m using a nearest-neighbor interpolation. Data from three national scale, 1x1km, gridded, soil property
maps (ACLEP, 2011) were used: (1) plant available water-holding capacity, calculated as the sum of the water-holding capacity of the A and B soil horizons (0 to 1m); (2) clay content (0 to 0.3m); and (3) soil bulk density (0 to 0.3m, spanning A and B horizons) as a measure of porosity. Foliage projective cover (FPC) was obtained from Danaher et al. (2004) although it was only available for one year prior to the study period. Pixels with
over 50% FPC (mostly dense tropical forest) were not included.

A k-means unsupervised clustering was used to classify meteorological data, soil properties, and FPC for each growing season. To ensure equal numerical weighting, all environmental data were normalized prior to clustering. The environmental data were then partitioned using unsupervised clustering (n=50, maximum iterations = 100, change
threshold = 0.05%, minimum of 1,000 pixels), which resulted in 50 clusters. These are referred to as UMD Land Capability Classes (UMDLCC). The pixels found within each homogeneous UMDLCC were examined using linear regression and Person correlation to determine if any underlying relationships remained between NPP and the environmental data used to create them. Only LCCs where the correlation was below 0.4 were included
in the final UMDLCC for that year. Pixels with correlations above 0.4 were reclassified. This procedure was repeated for each year.

Few maps exist that could be used for validation of the homogeneity of LCCs in the BDT. One such is the Grazing Land Management (GLM) Land Types (DPI&F, 2004; Whish, 2011) which classifies areas based on vegetation, soil, and terrain characteristics
to create types within which the response to grazing pressure is similar. Since the principles used to create GLM were similar to those of the UMDLCCs, an additional LNS was performed using GLM land types (GLMLCC). The vector GLM map was converted to a raster format at a 250x250m spatial resolution. GLM land types consisting of fewer than 1,000 pixels were removed, resulting in 50 GLMLCCs – the same number
of LCCs as the UMDLCC.  These were compared with the UMDLCC.

The two LCCs were compared using the mean square variance of their maximum NPP to determine the extent to which each reduced within-LCC variance and maximized between-LCC variance. Inter-annual wet season rainfall (Nov to Apr) was averaged throughout the BDT (Figure 2), and then compared with the two variance components of




both UMDLCC and GLMLCC. A paired t-test was used to determine whether there were significant differences in within-LCC and between-LCC variance in maximum NPP for the two LCCs.

A second comparison was made using the Vegetation Assets, States and Transitions (VAST) classification of Australia, version 2 (Lesslie et al. 2013). VAST is a national level map of changes to vegetation since European settlement, which began in 1750, showing the degree of anthropogenic modification of native vegetation until 2011. VAST uses the following classes: wilderness, biophysical naturalness, land use, land cover, and extent of native vegetation. There are four classes of increasing human

modification: 1-'modified', 2-'tranformed', 3-'replaced', and 4-'removed'. Areas without naturally occuring native vegetation are designated 5-'bare' and areas with no change as 0-'residual'.

Erosion is strongly linked to land degradation in drylands (Lal, 2003; Ravi et al., 2010), and this is the case in Australian rangelands (Bui et al., 2011; Dregne, 1995;

Gillieson et al., 1996; Webb et al., 2009). A datbase of erosion was used to better understand the nature of the degradation that was detected. Four environmental variables related to natural and human-related erosion processes were used: sediment load at 500x500m (NLWRA, 2002); soil erodibility, rainfall erosivity and hillslope erosion, each at 250x250m (Lu et al., 2001); and gully density at 500x500m (Hughes et al., 2001).

Gully density and sediment load were downscaled from their original spatial resolutions to 250x250m using a nearest-neighbor interpolation.

### 2.3 Measurement of NPP using satellite data

Moderate Resolution Imaging Spectrometer (MODIS) NPP data (MOD17A3) (Running et al., 2004) were obtained from the Land Processes Distributed Active Archive

Center satellite data archives (http://modis.gsfc.nasa.gov/data/; *accessed 06/05/2014*). These data have 1x1km resolution and so, to maintain the highest possible spatial resolution, the data were rescaled to 250x250m using coefficients of the regression of growing season 250x250m NDVI (MOD13Q1) on 1x1km, NDVI (MOD13A2).

LNS is spatially and temporally scale-dependent since the NPP in a pixel is the

sum of its finer scale components, similarly for the individual years in the 14 years that were studied may be different. Therefore, in this application, degradation at finer spatial and temporal scales than 250x250m and 14 years may have been missed, as would any pattern of LNS at finer scales (such as confinement of degradation to small, but repeated ridges in a distributary. While this might be a drawback for fine scale applications, such

as the effects of livestock congregation at water and gates, in the BDT, livestock management is normally applied to areas large enough to contain at least several 250x250m pixels. Other limitations for which there are no perfect solutions include the effect of gradients in environmental factors, such as meteorological variables, that are





dissected by the classification into arbitrary ranges. Pixels are more likely to be selected as reference sites if they are in the most favorable part of the gradient, often at the edge of LCC. While this effect is minimized using a large number of LCCs, it cannot be removed entirely. A warning situation would be if reference pixels were confined to one part of an LCC. In all of these cases, care is needed to review the LCCs using alternative sources such as high-resolution imagery that can provide visual warning. Additional limitations can arise if small features occur that are not large enough to be placed in a different LCC, also the situation where the entire LCC is degraded or entirely non-degraded. Various methods can be used to minimize these and other problems, but they cannot all be entirely prevented and in some cases the extent of the effect cannot be measured.

### 2.4 Local Net primary production scaling (LNS)

LNS values are the difference between each pixel and its reference NPP (Figure 3). It is therefore zero (equal to the reference NPP, i.e. not degraded) or negative (below the reference, i.e. degraded). The LNS values can be expressed as the actual reduction of NPP in gC m$^{-2}$ year$^{-1}$ or as a percentage of the reference. LNS was calculated for each year (2000-2013), producing 14 LNS maps, using both the UMDLCC and GLMLCC maps.

The potential, non-degraded reference NPP was obtained using the frequency distribution of NPP in each LCC (Figure 3). The 85$^{th}$ percentile was arbitrarily selected as the best estimator. Pixels with NPP higher than the reference, possibly caused by residual pixels with high NPP in areas that were not typical of the rest of the LCC, were omitted. A possible limitation of LNS is if no pixels are at their maximum; then the reference would be below the true potential. Masking rivers, open water, roads, human settlements, and other human land features not representative of the LCC minimized this effect, but it cannot be entirely eliminated and so interpretation of the results must take this into account.

LNS percent values were averaged from 2000 to 2013 to determine the mean NPP reduction for each pixel over the 14 years. To facilitate discussion, values that were ≤ -30% were arbitrarily classified as 'degraded'. All other pixels, those where LNS was between 0% and -29% were classified as 'non-degraded'. A time-series of annual LNS percent values for every pixel was used to identify significant ($\alpha<0.10$) inter-annual trends in LNS over the 14-years. Pixels were classified according to their trends into three categories: (1) no significant inter-annual trends ('no LNS trend'); (2) significant positive inter-annual trends ('positive LNS trend'); and (3) significant negative inter-annual trends ('negative LNS trend'). The trend classification was combined with the two levels of degradation to create six classes: (1) 'non-degraded and positive LNS trends', (2) 'non-degraded and no LNS trend', (3) 'non-degraded and negative LNS trend', (4) 'degraded and positive LNS trend', (5) 'degraded and no LNS trend', and (6) 'degraded and negative LNS trend'.





Spatial agreement between average LNS values and ecological indicators related to land condition (e.g. hillslope and gully erosion) or susceptibility to poor condition (e.g. rainfall erosivity and soil erodibility) were examined using Cohen's kappa (k) fuzzy numeric (Cohen, 1960). This elaboration of the simple kappa test includes 'near misses'
and allows for coincidences that occur by chance. Values range from 0.0 to 1.0 with increasing agreement. All kappa calculations were performed using the Map Comparison Kit (Visser and de Nijs, 2006).

## 3 Results

### 3.1 UMDLCC

The average number of pixels per UMDLCC varied each year from 3,182 ($0.01 \times 10^6$ km$^2$) in 2004 to 141,690 ($0.56 \times 10^6$ km$^2$) in 2013. Their locations differed each year owing to inter-annual differences in weather patterns. Approximately half were non-contiguous, interspersed between other LCCs, but generally in no more than two river basins. Most reference pixels were selected in more than one year and a small number
were selected in all years.

The inter-annual, between-LCC variance in reference NPP was higher for UMDLCC compared with GLMLCC. Conversely, within-LCC variance for UMDLCC was lower than for GLMLCC, indicating that the pixels selected as reference within UMDLCCs were more homogeneous than GLMLCC and more distinct between. A
20 paired t-test showed that these differences were significant (Table 1),

Inter-annual rainfall was significantly related to between-LCC and within-LCC variance in reference NPP for both LCCs (Figure 4), accounting for nearly equal proportions of within-LCC variance in reference NPP for UMDLCC and GLMLCC (Figure 4a), but between-LCC variance was better accounted for by UMDLCC (81%)
than for GLMLCC (66%; Figure 4b).

The comparison of UMDLCC and the VAST land classification, albeit based on different data and aims, provided an independent comparison. 35.8% of UMDLCC reference pixels were in the VAST 'residual' class that has, theoretically, been undisturbed since 1750. The remaining 64.2% were in classes with varying degrees of
30 vegetation changes from native pasture: 1-'modified' (29.6%); 2-'transformed' (19.2%); and 3-'replaced' (15.3%). The remaining reference sites, less than one percent, were in the 4-'removed' or 5-'bare' classes which may LCCs where all pixels were degraded, or have been caused by inadequate or inaccurate data used to create the LCCs, errors in the VAST classification, or a result of re-gridding VAST pixels from 1x1km to 250x250m
spatial resolution.





### 3.2 LNS

The -30% LNS percent value used to differentiate 'degraded' areas from 'non-degraded' areas was equivalent to an average annual reduction in NPP of -169.6 gC m$^{-2}$ year$^{-1}$ (standard deviation=25.5). Between 2000 and 2013 the average annual LNS across the entire BDT, including both 'degraded' and 'non-degraded' areas, was -2.14 MgC m$^{-2}$ year$^{-1}$ (Table 2). The average reduction in 'degraded' areas was more than twice that in the 'non-degraded' areas and the LNS of the positive LNS trend class was lower than the negative and no LNS trends.

The sum of LNS values for entire class, as opposed to LNS per unit area revealed how the importance the size of each class in contributing to the overall reduction in NPP. The 'degraded' class had a total reduction in NPP of -1.1 GgC from 2000 to 2013 and occupied 1.46x10$^6$ km$^2$ (Table 3). The larger area occupied by the 'non-degraded' class resulted in a greater total reduction in NPP (-1.9 GgC; Table 3), although much less severe reduction in NPP per unit area (Table 2). In the same way, non-degraded areas with no LNS trend had by far the greatest total reduction in NPP owing to the large area occupied by this class.

The majority of degraded pixels had LNS values between -30% and -49%, with only a small proportion below -50%. The largest number of the non-degraded pixels were in the -10% to -29% LNS classes. For the degraded pixels, the average LNS in NPP units was less than half that of the non-degraded pixels (Tables 4 & 5). Similarly, reductions in NPP as a percent of the reference were smaller (more severe) for degraded than non-degraded pixels.

### 3.3 Spatial variation in LNS

The extent of 'degraded' and 'non-degraded' areas varied between the six major river basins (Tables 4 & 5). Two of these, Belyando and Suttor, comprised 67% of all 'degraded' areas in the entire BDT while the Bowen Broken Bogie had the lowest (2%) (Table 4). Despite being the first and third largest basins in the BDT ('degraded' plus 'non-degraded' pixels) the Upper Burdekin and Cape Campaspe had only the third and fourth most 'degraded' pixels (Table 4), respectively. However, 'non-degraded' area decreased with decreasing size of each river basin (Table 5).

The severity of reductions in NPP, indicated by the average LNS, varied surprisingly little between river basins (Tables 4 & 5). The most severely degraded were in the Lower Burdekin, Bowen Broken Bogie, and Upper Burdekin (Table 4). The Upper Burdekin also had the most severe reductions of non-degraded pixels (Table 5). The Belyando and Cape Campaspe had the least severe reductions in NPP of degraded and non-degraded pixels, respectively. The average LNS and its percentage of the reference



NPP for degraded and non-degraded pixels, however, were all within one standard deviation; suggesting that the reductions in NPP for each river basin did not differ substantially.

Among degraded areas there was evidence of managed grazing, including abrupt differences in LNS along station boundaries (Figure 5b), but there were also gradients of LNS within some stations (Figure 5c), and others with low LNS spread across boundaries (Figure 5d). Other areas with evidence of management included forest clearing (Figure 5e) near station boundaries. There were also locations classified as degraded with little evidence of direct grazing management such as between the drainage lines of streams (Figure 5f).

### 3.4 Inter-annual trends in LNS

Across the entire BDT there was substantial inter-annual variation in LNS, particularly in areas with low values (Figure 6a). In years with high rainfall (e.g. 2000, 2008, 2009 and 2011) compared with low rainfall (e.g. 2003, 2005 and 2013), there were fewer pixels with low LNS, but the severity of reductions was greater. In areas with little topographic variation, such as the central BDT, there was more spatial variation in low values between years. Positive trends were found predominately in the western and southern Upper Burdekin and southern Belyando basins. Negative trends were most common in the northern Belyando, central Upper Burdekin, and southern Suttor river basins. 79.4% of the BDT had no significant trend in LNS.

The magnitudes of negative and positive inter-annual trends in LNS varied substantially between river basins (Fig. 6b, Tables 6 & 7). The Suttor had by far the lowest negative trends (but the largest standard deviation; Table 6). The Upper Burdekin and Cape Campaspe had the least negative trends (Table 6). Positive trends were highest in the Bowen Broken Bogie and lowest in the Belyando (Table 7).

Some patches of positive and negative LNS trends were found in large areas that spanned multiple river basins (Figure 6b). These may have been a result of environmental conditions (e.g. low rainfall, soil properties) in some combination other than that used to create the LCCs, or a single variable not used in the classification, that crosses the LCC boundaries, for example, more friable soils.

There were strong contrasts in the average LNS of the negative and positive trend classes between river basins (Tables 6 & 7). The average LNS of negative trends in the Suttor was nearly twice that of the Upper Burdekin. The Suttor river basin had most severe LNS reductions in the negative trend class (Table 6). On average, for negative trends, the Bowen Broken Bogie, Upper and Lower Burdekin had the least severe reductions in NPP while the most severe were in the southern river basins: Belyando, Cape Campaspe, and Suttor (Table 6). Surprisingly, the Belyando had less severe





reductions in NPP in areas with negative trends (Table 6) than in areas with positive trends (Tables 7). In the Belyando, the percent LNS for positive trends were less than -30%, suggesting that numerous low LNS values were found among positive trends.

### 3.5 Comparisons of LNS and environmental characteristics

For the entire BDT, the overall spatial distribution of annual hillslope erosion was strongly correlated ($k = 0.7$) with LNS. Other environmental variables indicative of degradation (gully density, rainfall erosivity, and sediment load) were also high overall, ($k = 0.6$). For individual pixels, maps of correlation revealed strong regional differences (Figure 7). The Suttor had the greatest spatial agreement between LNS and each

environmental variable, while the Bowen Broken Bogie had the least. Strong agreement between annual hillslope erosion and LNS were occurred in throughout the BDT (Figure 7a), particularly in the central Upper Burdekin, Cape Campaspe, Suttor and Belyando. The spatial agreement between LNS and gully density (Figure 7b) were largely similar to that of LNS and hillslope erosion except the presence of large clusters of low kappa

values in the northern basins. The spatial pattern in kappa values for LNS and rainfall erosivity (Figure 7c) and sediment load (Figure 7d) resembled rainfall gradients in the region, northeast to southwest.

        VAST classes were generally correlated with LNS (Table 8). The average LNS declined with increasing human modification. 'Removed' and 'bare' had the lowest

average LNS of any VAST class. The only negative trend was in the 'bare' class while 'removed' had the largest positive trend. Inter-annual trends in LNS further differentiated the two classes; 'bare' had the only negative trend while 'removed' had the largest positive trend, which may be a result of the strongest regrowth from the lowest starting value of all the classes.

## 4 Discussion

### 4.1 Land capability classification (LCC) and local NPP scaling (LNS)

        The basis of selection of the reference NPP and detection of anthropogenic reductions in LNS is the classification of the landscape into uniform units (LCCs) with respect to the environmental factors that affect NPP. The procedure was generally

successful in creation of classes of environmentally uniform pixels, differing only in the long-term degree of degradation. The same reference sites were frequently selected in multiple, sometimes consecutive, years for the 14 years included in the study and therefore potentially for a longer term. This indicates that degradation, as detected with LNS, were sites that were persistently below the potential, not simply subject to some

short-term environmental deficiency, such a single-year with spatially patchy lower rainfall. The value of incorporating inter-annual variation of precipitation in the classification rather than a climatological average is illustrated by the comparison of





GLM. UMDLCC proved better able to minimize within-LCC variance while also maximizing the between-LCC variance (Table 1, Figure 4a & Figure 4b). The large numbers of UMD reference sites that fell in the VAST 'residual' class and the larger reductions in NPP in VAST classes with higher levels of human modification, offer

further evidence of the reliability of the UMDLCC classification (Table 8). Furthermore, the spatial coincidence of differences in management with differences in LNS found by visual inspection of high resolution imagery suggests that the procedure was able to distinguish regional, anthropogenic land degradation from natural variation in environmental factors.

Nevertheless, undetected errors may arise in the classification process, some of which are noted in the Methods section. Changes in land cover during the study period are unlikely to have caused errors since the rates of pasture clearing decreased dramatically throughout the Burdekin region from 1988 to 2002 and remained relatively low during the study period (2000 to 2012, (DSITIA, 2014). A more fundamental

problem might arise because the classification procedure did not allow for any interactions between environmental factors in different parts of the study area. A possible example of this from the BDT is the location of the largest spatial variations in LNS and its inter-annual trends near the coastline (e.g. Lower Burdekin and Bowen Broken Bogie) where rainfall is highest. This is an example of a drawback of statistical classification

which can only account for additive effects of the environment whereas, for example, moisture availability can alter the response of production to management (Ibrahim et al., 2015), maybe non-linearly. This points to an advantage of replacing the statistical derivation of LCCs with a process-based model that can convolve the environmental factors in realistic mechanisms. Such a model run in "potential" mode, which is without

any anthropogenic effects, could create a reference NPP for each pixel. At the present time, however, the environmental variables and parameters needed for a useful process model are only rarely available.

### 4.2 Extent of degradation of NPP in the BDT

Across the entire BDT region, from 2000 to 2013, the average annual reduction in
NPP below the reference was 2.14 MgC m$^{-2}$ year$^{-1}$ (Table 2). The average LNS in the non-degraded class (arbitrarily set at LNS between 0 and -29%) was -97.5 gC m$^{-2}$ year$^{-1}$ and the degraded class (LNS <30%) -209.1 gC m$^{-2}$ year$^{-1}$ (Tables 4 & 5). However, owing to the greater area of 'non-degraded' land in the entire BDT (80.3%) compared to the 'degraded' class (19.7%) (Table 2), the total NPP reduction in non-degraded areas was
actually greater. Reductions in NPP, as indicated by low LNS, affect the carbon pool in several ways: by reduced rates of sequestration (Dregne, 1983); by reduction in biomass of live and dead vegetation; by loss of soil organic matter (Burke et al., 1995; Smith et al., 2012; Su et al., 2005); or by a shortened growing season, for example among introduced, less-adapted, pasture species (Falge et al., 2002a; Falge et al., 2002b). The



large reduction in NPP found here is in agreement with reports of episodes of widespread land degradation occurring in the BDT (McKeon et al., 2004a; Smith et al., 2007; Stone et al., 2007).

Overall, positive temporal trends in LNS were twice as common as negative trends (Tables 6 & 7). The 'Non-degraded with no trend' class had the largest total area (65%). This class was widespread in every river basin, indicating that most of the BDT region was not affected by severe degradation. In other areas, for example in Belyando and Bowen Broken Bogie, the average LNS of 'Degraded with positive trends' areas suggests that significant areas were recovering from earlier degradation (Table 7).

Nevertheless, some areas were degrading between 2000 and 2013 and in some, their negative trends intensified through the study period, as indicated by the extent of the 'Degraded with negative trends' class (Table 1). Areas classified as 'Degraded with negative trends' occupied 24.7% of the entire BDT –candidate areas for actions to reverse or at least arrest the trend. There were a few instances of 'Degraded with no trends',

presumably sites in a state of long-term, maybe permanent, irreversible degradation or approaching this state. Permanent degradation is a serious condition since it is generally reversible only with intensive remediation (Prince, 2002; Reynolds et al., 2007) which often costs more than the value of the restored land, however, there were a few areas of 'Degraded with positive trends' which may be examples of land that has been

rehabilitated.

### 4.3 Anthropogenic and environmental degradation

While substantial reductions in NPP were found across the BDT region as whole, there was considerable variation between river basins. The link between anthropogenic disturbance and rates of degradation (detected here by low LNS) has been noted by Hill

et al. (2005) and Kairis et al. (2015) and specifically in the BDT by McKeon et al. (2009). Independent evidence for anthropogenesis presented here includes correlation with the VAST map which, although not a map of vegetation degradation, does distinguish varying degrees of human-related modification of native vegetation (Thackway and Lesslie, 2005). The strong correlation of ranks of average LNS and the

VAST classes (Table 8) is evidence that LNS was able to separate human-related degradation from natural variation, at least up to the end of the period of time used for the VAST map (2011). In addition, there was qualitative evidence from visual inspection of high resolution remotely-sensed imagery, such as abrupt differences across station boundaries (e.g. Figure 5b & Figure 5c) and coincidences of visible disturbance around

livestock water points. The relationship between degradation, accelerated rates of erosion, and reduced vegetation cover is well-known (Lal, 2001) and erosion is the most widespread and recognizable characteristics of land degradation (Ravi et al., 2010), also a primary impact on loss of soil carbon (Rajan et al., 2010). In the present study, there was a strong overall correlation of average LNS with hillslope erosion and gully density





(Figure 7). In the BDT others have linked erosion with poor grazing management (Bartley et al., 2006) and unsustainable agricultural production (Montgomery, 2007).

Assigning causal relationships to land degradation and natural or anthropogenic factors is difficult due to the close coupling between humans and their environment (Reynolds et al., 2007). The LNS procedure offers one approach that attempts to isolate actual degradation of NPP from less favorable environmental conditions. However, without additional data on land usage, such as livestock numbers and management practices, the causes of the reductions by human-related activities are hard to determine (Bastin et al., 2012). The most commonly-cited management practices to reduce degradation are reduction in domestic livestock, reduction of feral herbivores, removal of watering points (Bastin et al., 2012; Fensham and Fairfax, 2008; Silcock and Fensham, 2013), fallowing (Bastin et al., 2012; Bastin et al., 1993), or by encouraging vegetation that is particularly resistant to overgrazing or able to recover quickly after intense grazing (Bastin et al., 2012; McKeon et al., 2004b; Smith et al., 2007). Additional data are needed to interpret low LNS, particularly with field observation.

Given the extremely large areas of provincial, national, regional and global degradation that are frequently stated (Bai et al., 2008; Bridges and Oldeman, 1999; Kassas, 1995; Oldeman, 1994; UNEP, 1997; Zika and Erb, 2009) and the far-reaching effects of degradation on human livelihoods (Adeel, 2008; UNCCD, 1994), rigorous, quantitative and objective measurements are urgently needed. While reduction of NPP is a single type of degradation, it is a quantitative measure of the outcome of most forms of degradation relevant to human needs - but not all (e.g. loss of palatable species with no change in NPP) (Asner and Heidebrecht, 2005). The widespread occurrence of degradation and its anthropogenic causes and effects require measurements having the large-area coverage and high spatial resolution provided by remote sensing, despite their limitations. LNS is founded on the concept of comparison of the actual conditions with their potential. As noted, there are several weaknesses in the technique that may affect the validity of the results, nevertheless, the fundamental concept of reduction from an explicit standard remains. There remains a need for improvements in detection of appropriate reference standards, either by local scaling as in LNS or some other method.

The objectives of the many initiatives to arrest and remediate degradation have been summarized in the concept of Zero Net Land Degradation (ZNLD) (Stavi and Lal, 2015). ZNLD seeks to slow current rates of degradation such that the rates of land rehabilitation are, at the very least, equivalent to rates of deterioration (Lal et al., 2012), locally or elsewhere. Achievement of ZNLD depends on comprehensive monitoring to identify land states and trends of degradation. The study presented here used one approach to such regional assessment. While the feasibility of global land degradation neutrality has been debated (Grainger, 2015), the BDT is an example of a region that has seen a reversal of an overall trend toward degradation in productivity.





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

Box 17, 3300 Aa Dordrecht, Netherlands, 2008.

Asner, G. P. and Heidebrecht, K. B.: Desertification alters regional ecosystem-climate interactions, Global Change Biology, 11, 182-194, 2005.

Bai, Z. G., Dent, D. L., Olsson, L., and Schaepman, M. E.: Proxy global assessment of land degradation, Soil Use and Management, 24, 223-234, 2008.

Bartley, R., Roth, C. H., Ludwig, J., McJannet, D., Liedloff, A., Corfield, J., Hawdon, A., and Abbott, B.: Runoff and erosion from Australia's tropical semi-arid rangelands: influence of ground cover for differing space and time scales, Hydrological Processes, 20, 2006.

Bastin, G., Scarth, P., Chewings, V., Sparrow, A., Denham, R., Schmidt, M., O'Reagain,
P., Shepherd, R., and Abbott, B.: Separating grazing and rainfall effects at regional scale using remote sensing imagery: A dynamic reference-cover method, Remote Sensing of Environment, 121, 443-457, 2012.

Bastin, G. N., Pickup, G., Chewings, V. H., and Pearce, G.: Land degradation assessment in central Australia using grazing gradient method, The Rangeland Journal, 15,
190-216, 1993.

Beringer, J., Hutley, L. B., Tapper, N. J., and Cernusak, L. A.: Savanna fires and their impact on net ecosystem productivity in North Australia, Global Change Biology, 13, 990-1004, 2007.

Boer, M. and Smith, M. S.: A plant functional approach to the prediction of changes in
Australian rangeland vegetation under grazing and fire, Journal of Vegetation Science, 14, 333-344, 2003.

Boer, M. M. and Puigdefabregas, J.: Assessment of dryland condition using spatial anomalies of vegetation index values, International Journal of Remote Sensing, 26, 4045-4065, 2005.

Bridges, E. M. and Oldeman, L. R.: Global assessment of human-induced soil degradation, Arid Soil Research and Rehabilitation, 13, 1999.





Budde, M. E., Tappan, G., Rowland, J., Lewis, J., and Tieszen, L. L.: Assessing land cover performance in Senegal, West Africa using 1-km integrated NDVI and local variance analysis, Journal of Arid Environments, 59, 481-498, 2004.

Bui, E. N., Hancock, G. J., and Wilkinson, S. N.: 'Tolerable' hillslope soil erosion rates in Australia: Linking science and policy, Agriculture Ecosystems & Environment, 144, 2011.

Burke, I. C., Lauenroth, W. K., and Coffin, D. P.: SOIL ORGANIC-MATTER RECOVERY IN SEMIARID GRASSLANDS - IMPLICATIONS FOR THE CONSERVATION RESERVE PROGRAM, Ecol. Appl., 5, 793-801, 1995.

Cohen, J.: A coefficient of agreement for nominal scales, Educational and Psychological Measurement, 20, 37-46, 1960.

Danaher, T., Armston, J., Collett, L., and ieee: A regression model approach for mapping woody foliage projective cover using landsat imagery in Queensland, Australia, Igarss 2004: Ieee International Geoscience and Remote Sensing Symposium Proceedings, Vols 1-7: Science for Society: Exploring and Managing a Changing Planet, 2004. 523-527, 2004.

DPI&F: Stocktake: Balancing supply and demand., Department of Primary Industries and Fisheries, 2004.

Dregne, H. E.: Desertification of arid lands. In: Physics of desertification, El-Baz, F. and Hassan, M. H. A. (Eds.), Harwood Academic Publishers, Chur, Switzerland ; New York, 1983.

Dregne, H. E.: EROSION AND SOIL PRODUCTIVITY IN AUSTRALIA AND NEW-ZEALAND, Land Degradation and Rehabilitation, 6, 1995.

DSITIA: Land cover change in Queensland 2011–12: a Statewide Landcover and Trees Study (SLATS) Report, 2014., Department of Science, Information Technology, Innovation and the Arts, Brisbane., Department of Science, Information Technology, Innovation and the Arts, 2014.

Falge, E., Baldocchi, D., Tenhunen, J., Aubinet, M., Bakwin, P., Berbigier, P., Bernhofer, C., Burba, G., Clement, R., Davis, K. J., Elbers, J. A., Goldstein, A. H., Grelle, A., Granier, A., Guomundsson, J., Hollinger, D., Kowalski, A. S., Katul, G., Law, B. E., Malhi, Y., Meyers, T., Monson, R. K., Munger, J. W., Oechel, W., Paw, K. T., Pilegaard, K., Rannik, U., Rebmann, C., Suyker, A., Valentini, R., Wilson, K., and Wofsy, S.: Seasonality of ecosystem respiration and gross primary production as derived from FLUXNET measurements, Agricultural and Forest Meteorology, 113, 53-74, 2002a.

Falge, E., Tenhunen, J., Baldocchi, D., Aubinet, M., Bakwin, P., Berbigier, P., Bernhofer, C., Bonnefond, J. M., Burba, G., Clement, R., Davis, K. J., Elbers, J. A., Falk, M., Goldstein, A. H., Grelle, A., Granier, A., Grunwald, T., Gudmundsson, J., Hollinger, D., Janssens, I. A., Keronen, P., Kowalski, A. S., Katul, G., Law, B. E.,





Malhi, Y., Meyers, T., Monson, R. K., Moors, E., Munger, J. W., Oechel, W., U, K. T. P., Pilegaard, K., Rannik, U., Rebmann, C., Suyker, A., Thorgeirsson, H., Tirone, G., Turnipseed, A., Wilson, K., and Wofsy, S.: Phase and amplitude of ecosystem carbon release and uptake potentials as derived from FLUXNET measurements, Agricultural and Forest Meteorology, 113, 75-95, 2002b.

Fensham, R. J. and Fairfax, R. J.: Water-remoteness for grazing relief in Australian arid-lands, Biological Conservation, 141, 1447-1460, 2008.

Gifford, R. M.: Carbon sequestration in Australian grasslands: policy and technical issues. In: Integrated Crop Management Grassland carbon sequestration: management, policy and economics. Proceedings of the Workshop on the role of grassland carbon sequestration in the mitigation of climate change, Abberton, M., Conant, R., and Batello, C. (Eds.), Food and Agriculture Organization of the United Nations (FAO), Rome, April 2009, 2010.

Gillieson, D., Wallbrink, P., and Cochrane, A.: Vegetation change, erosion risk and land management on the Nullarbor Plain, Australia, Environmental geology, 28, 145-153, 1996.

Grainger, A.: Is Land Degradation Neutrality feasible in dry areas?, Journal of Arid Environments, 112, 14-24, 2015.

Hill, M. J., Roxburgh, S. H., Carter, J. O., and McKeon, G. M.: Vegetation state change and consequent carbon dynamics in savanna woodlands of Australia in response to grazing, drought and fire: a scenario approach using 113 years of synthetic annual fire and grassland growth, Australian Journal of Botany, 53, 715-739, 2005.

Hughes, A. O., Prosser, I. P., Stevenson, J., Scott, A., Lu, H., Gallant, J., and Moran, C. J.: Gully Erosion Mapping for the National Land and Water Resources Audit. In: Technical Report 26/01, Water, C. L. a. (Ed.), Canberra, Australia, 2001.

Hutley, L. B., O'Grady, A. P., and Eamus, D.: Evapotranspiration from eucalypt open-forest savanna of northern Australia, Functional Ecology, 14, 183-194, 2000.

Ibrahim, Y. Z., Balzter, H., Kaduk, J., and Tucker, C. J.: Land Degradation Assessment Using Residual Trend Analysis of GIMMS NDVI3g, Soil Moisture and Rainfall in Sub-Saharan West Africa from 1982 to 2012, Remote Sensing, 7, 5471-5494, 2015.

Jones, D. A., Wang, W., and Fawcett, R.: High-quality spatial climate data-sets for Australia, Australian Meteorological and Oceanographic Journal, 58, 2009.

Kairis, O., Karavitis, C., Salvati, L., Kounalaki, A., and Kosmas, K.: Exploring the Impact of Overgrazing on Soil Erosion and Land Degradation in a Dry Mediterranean Agro-Forest Landscape (Crete, Greece), Arid Land Research and Management, 29, 360-374, 2015.



Kassas, M.: Desertification - a general-review, Journal of Arid Environments, 30, 115-128, 1995.

Lal, R.: Soil degradation by erosion, Land Degradation & Development, 12, 519-539, 2001.

Lal, R.: Soil erosion and the global carbon budget, Environment International, 29, 2003.

Lal, R., Safriel, U., and Boer, B.: Zero Net Land Degradation: A New Sustainable Development Goal for Rio+ 20, UNCCD, 2012.

Lu, H., Gallant, J., Processer, I. P., Moran, C., and Priestley, G.: Predication of Sheet and Rill Erosion over the Australian Continent, Incorporating Monthly Soil Loss
Distribution. 13/01, C. L. a. W. T. R. (Ed.), National Land & Water Resource Audit, Canberra, Australia, 2001.

McKeon, G., Hall, W., Henry, B., Stone, G., and Watson, I. (Eds.): Pasture Degradation and Recovery in Australia's Rangelands: Learning from History, Department of Natural Resources, Mines and Energy, 2004a.

McKeon, G., Hall, W., Henry, B., Stone, G., and Watson, I.: Pasture Degradation and Recovery in Australia's Rangelands: Learning from History, Department of Natural Resources Mines and Energy Queensland, Brisbane, 2004b.

McKeon, G. M., Stone, G. S., Syktus, J. I., Carter, J. O., Flood, N. R., Ahrens, D. G., Bruget, D. N., Chilcott, C. R., Cobon, D. H., Cowley, R. A., Crimp, S. J., Fraser,
G. W., Howden, S. M., Johnston, P. W., Ryan, J. G., Stokes, C. J., and Day, K. A.: Climate change impacts on northern Australian rangeland livestock carrying capacity: a review of issues, Rangeland Journal, 31, 1-29, 2009.

Mellick and Hanlon, H.: The Burdekin Dry Tropics Natural Resources Management Plan (2005-2010), Burdekin Dry Tropics Board, 2005, 2005.

Montgomery, D. R.: Soil erosion and agricultural sustainability, PNAS, 104, 13268-13272, 2007.

NLWRA: Catchment, river and estuary condition in Australia : a summary of the National Land and Water Resources Audit's Australian catchment, river and estuary assessment 2002., Audit, N. L. a. W. R. (Ed.), Natural Heritage Trust (Australia) & National Land & Water Resources Audit (Program : Australia),
Turner, A.C.T., 2002.

Nusser, S. M. and Goebel, J. J.: The National Resources Inventory: A long-term multi-resource monitoring programme, Environmental and Ecological Statistics, 4, 181-204, 1997.

O'Connor, T. G., Haines, L. M., and Snyman, H. A.: Influence of precipitation and species composition on phytomass of a semi-arid African grassland, Journal of Ecology, 89, 850-860, 2001.





Oldeman, L. R.: The global extent of soil degradation, 1994.

Petheram, C., McMahon, T. A., and Peel, M. C.: Flow characteristics of rivers in northern Australia: Implications for development, Journal of Hydrology, 357, 93-111, 2008.

5    Pickup, G.: Desertification and climate change - the Australian perspective, Climate Research, 11, 1998.

Pickup, G.: Estimating the effects of land degradation and rainfall variation on productivity in rangelands: An approach using remote sensing and models of grazing and herbage dynamics, Journal of Applied Ecology, 33, 1996.

10   Prince, S. D.: Mapping desertification in southern Africa. In: Land Change Science: Observing, Monitoring, and Understanding Trajectories of Change on the Earth's Surface, Gutman, G., Janetos, A., and Justice, C. O. (Eds.), Kluwer, Dordrecht, Netherlands, 2004.

Prince, S. D.: Spatial and temporal scales of measurement of desertification. Global desertification: do humans create deserts? , Reynolds, M. S.-S. a. J. F. (Ed.), Dahlem University Press, Berlin, 2002.

Prince, S. D.: Where Does Desertification Occur? Mapping Dryland Degradation at Regional to Global Scales. In: The End of Desertification? Disrupting Environmental Change in Drylands, Behnke, R. and Matimore, M. (Eds.), Springer, 2016.

Prince, S. D., Becker-Reshef, I., and Rishmawi, K.: Detection and mapping of long-term land degradation using local net production scaling: Application to Zimbabwe, Remote Sensing of Environment, 113, 1046-1057, 2009.

Prince, S. D., Wessels, K. J., Tucker, C. J., and Nicholson, S. E.: Desertification in the Sahel: a reinterpretation of a reinterpretation, Global Change Biology, 13, 1308-1313, 2007.

Rajan, K., Natarajan, A., Kumar, K. S. A., Badrinath, M. S., and Gowda, R. C.: Soil organic carbon - the most reliable indicator for monitoring land degradation by soil erosion, Current Science, 99, 2010.

30   Ravi, S., Breshears, D. D., Huxman, T. E., and D'Odorico, P.: Land degradation in drylands: Interactions among hydrologic-aeolian erosion and vegetation dynamics, Geomorphology, 116, 2010.

Reynolds, J. F., Stafford Smith, D. M., Lambin, E. F., Turner, B. L., Mortimore, M., Batterbury, S. P. J., Downing, T. E., Dowlatabadi, H., Fernandez, R. J., Herrick, J. E., Huber-Sannwald, E., Jiang, H., Leemans, R., Lynam, T., Maestre, F. T., Ayarza, M., and Walker, B.: Global desertification: Building a science for dryland development, Science, 316, 847-851, 2007.



Running, S. W., Nemani, R. R., Heinsch, F. A., Zhao, M. S., Reeves, M., and Hashimoto, H.: A continuous satellite-derived measure of global terrestrial primary production, Bioscience, 54, 547-560, 2004.

Rustomji, P., Bennett, N., and Chiew, F.: Flood variability east of Australia's Great Dividing Range, Journal of Hydrology, 374, 196-208, 2009.

Safriel, U.: The Assessment of Global Trends in Land Degradation, Climate and Land Degradation, 2007. 1-38, 2007.

Safriel, U. and Adeel, Z.: Dryland Systems, Millennium Ecosystem Assessment: Ecosystems and Human Well-being: Current State and Trends, Washington, DC, 638, 2005.

Silcock, J. L. and Fensham, R. J.: Arid vegetation in disequilibrium with livestock grazing: Evidence from long-term exclosures, Austral Ecology, 38, 57-65, 2013.

Smith, D. M. S., McKeon, G. M., Watson, I. W., Henry, B. K., Stone, G. S., Hall, W. B., and Howden, S. M.: Learning from episodes of degradation and recovery in variable Australian rangelands, Proceedings of the National Academy of Sciences of the United States of America, 104, 20690-20695, 2007.

Smith, J. G., Eldridge, D. J., and Throop, H. L.: Landform and vegetation patch type moderate the effects of grazing-induced disturbance on carbon and nitrogen pools in a semi-arid woodland, Plant and Soil, 360, 405-419, 2012.

Stavi, I. and Lal, R.: Achieving Zero Net Land Degradation: Challenges and opportunities, Journal of Arid Environments, 112, 44-51, 2015.

Stoms, D. M. and Hargrove, W. W.: Potential NDVI as a baseline for monitoring ecosystem functioning, International Journal of Remote Sensing, 21, 401-407, 2000.

Stone, G., Dorine Bruget, John Carter, Robert Hassett, Greg McKeon, and David Rayner: Land: Pasture production and condition, Department of Natural Resources and Water, Queensland Government, 2007.

Su, Y. Z., Li, Y. L., Cui, H. Y., and Zhao, W. Z.: Influences of continuous grazing and livestock exclusion on soil properties in a degraded sandy grassland, Inner Mongolia, northern China, Catena, 59, 267-278, 2005.

Thackway, R. and Lesslie, R.: Vegetation Assets, States and Transitions (VAST):accounting for vegetation condition in the Australian landscape, Bureau of Rural Sciences, Canberra, 2005.

UNCCD: UNCCD Secretariat Policy Brief on Zero Net Land Degradation – a Sustainable Development Goal for Rio+20, United Nations Convention to Combat Desertification (UNCCD), Bonn, Germany, 2012.



UNCCD: United Nations convention to combat desertification in countries experiencing serious drought and/or desertification, particularly in Africa, United Nations General Assembly, New York, 1994.

UNEP: World Atlas of Desertification, Arnold & Wiley, on behalf of UNEP, London & New York, 1997.

Visser, H. and de Nijs, T.: The Map Comparison Kit, Environmental Modelling & Software, 21, 346-358, 2006.

Walker, B. H. and Janssen, M. A.: Rangelands, pastoralists and governments: interlinked systems of people and nature, Philosophical Transactions of the Royal Society of London Series B-Biological Sciences, 357, 719-725, 2002.

Webb, N. P., Phinn, S. R., and McGowan, H. A.: Visual assessment of the Australian Land Erodibility Model, Journal of Arid Environments, 73, 678-682, 2009.

Wessels, K. J., Prince, S. D., Malherbe, J., Small, J., Frost, P. E., and VanZyl, D.: Can human-induced land degradation be distinguished from the effects of rainfall variability? A case study in South Africa, Journal of Arid Environments, 68, 2007.

Wessels, K. J., Prince, S. D., and Reshef, I.: Mapping land degradation by comparison of vegetation production to spatially derived estimates of potential production, Journal of Arid Environments, 72, 1940-1949, 2008.

Weymouth, G., Mills, G. A., Jones, D., Ebert, E. E., and Manton, M. J.: A continental-scale daily rainfall analysis system, Australian Meteorological Magazine, 48, 1999.

Whish, G.: Land types of Queensland. Team, G. L. M. W. (Ed.), Department of Employment, Economic Development and Innovation, Brisbane, 2011.

Zika, M. and Erb, K. H.: The global loss of net primary production resulting from human-induced soil degradation in drylands, Ecological Economics, 69, 310-318, 2009.



**Table 1.** Mean, standard deviation, and t-test of mean square variance of reference NPP (gC m$^{-2}$ year$^{-1}$) for UMDLCC and GLMLCC, partitioned into between-LCC and within-LCC.

| Mean square variance | UMDLCC | | GLMLCC | | Significance level |
|---|---|---|---|---|---|
| | Mean | Std deviation | Mean | Std deviation | |
| Between LCCs | $4.15 \times 10^8$ | $1.1 \times 10^8$ | $1.76 \times 10^8$ | $0.5 \times 10^8$ | $t_{13}=12.6$ $p<0.0001$ |
| Within LCCs | $4.38 \times 10^4$ | $1.1 \times 10^4$ | $7.71 \times 10^4$ | $2.3 \times 10^4$ | $t_{13} = 9.6$ $p<0.0001$ |





**Table 2.** Average LNS (Mg C m$^{-2}$ year$^{-1}$) in the Burdekin Dry Tropics (BDT) region for all six combinations of degraded and non-degraded LNS conditions and three inter-annual LNS trends – no trend, positive and negative trends. The percentage of BDT area in each land condition is shown in parentheses.

| Trend category | Degradation condition (Mg C m$^{-2}$ year$^{-1}$) | | |
|---|---|---|---|
| | Non-degraded LNS | Degraded LNS | Average |
| No LNS trend | -1.70 (65.3%) | -3.85 (14.1%) | -2.08 (79.4%) |
| Positive LNS trend | -1.90 (10.0%) | -3.95 (3.6%) | -2.44 (13.6%) |
| Negative LNS trend | -1.48 (5.0%) | -4.14 (2.0%) | -2.24 (7.0%) |
| Average | -1.71 (80.3%) | -3.90 (19.7%) | -2.14 (100%) |





**Table 3.** Area and percentage of Burdekin Dry Tropics in each LNS range.

| Degradation Condition | LNS Range | Total area (km²) and percent of BDT | Total reduction in NPP (GgC) |
|---|---|---|---|
| Non-degraded | 0 to -9% | 1.12x10⁶ (15.8%) | -1.9 (80.3%) |
| | -10 to -19% | 2.40x10⁶ (33.9%) | |
| | -20 to -29% | 2.10x10⁶ (29.6%) | |
| Degraded | -30 to -39% | 0.96x10⁶ (13.6%) | -1.1 (19.7%) |
| | -40 to -49% | 0.35x10⁶ (5.0%) | |
| | -50 to -59% | 0.10x10⁶ (1.5%) | |
| | -60 to -69% | 0.03x10⁶ (0.4%) | |
| | -70 to -79% | 0.01x10⁶ (0.1%) | |
| | < -80 | 0.00x10⁶ (<0.0%) | |



**Table 4.** Degraded LNS class. Area, severity, and variation in LNS and LNS percent. sd – standard deviation

| River basin (in decreasing order of area) | Total area in km² and percent of the class | Average LNS in gC m⁻² year⁻¹ | Average LNS as a percentage of reference NPP |
|---|---|---|---|
| Upper Burdekin | $2.28 \times 10^5$ (16%) | -225.3 (sd=42.8) | -36% (sd=6) |
| Belyando | $6.60 \times 10^5$ (45%) | -200.2 (sd=47.5) | -40% (sd=8) |
| Cape Campaspe | $2.05 \times 10^5$ (14%) | -205.3 (sd=45.6) | -39% (sd=9) |
| Suttor | $3.17 \times 10^5$ (22%) | -215.3 (sd=52.0) | -40% (sd=9) |
| Bowen Broken Bogie | $0.33 \times 10^5$ (2%) | -225.7 (sd=45.7) | -36% (sd=7) |
| Lower Burdekin | $0.25 \times 10^5$ (2%) | -226.7 (sd=55.1) | -38% (sd=8) |
| Entire BDT region | $14.79 \times 10^5$ (100%) | -209.1 (sd=48.7) | -39% (sd=8) |



**Table 5.** Non-degraded LNS class. Area, severity, and variation in LNS and LNS percent. sd – standard deviation

| River basin (in decreasing order of area) | Total area in km² and percent of the class | Average LNS in gC m⁻² year⁻¹ | Average LNS as a percentage of reference NPP |
|---|---|---|---|
| Upper Burdekin | 20.34x10⁵ (34%) | -105.3 (sd=45.0) | -17% (sd=7) |
| Belyando | 14.15x10⁵ (24%) | -92.2 (sd=39.2) | -18% (sd=7) |
| Cape Campaspe | 9.74x10⁵ (16%) | -88.3 (sd=41.2) | -16% (sd=8) |
| Suttor | 7.57x10⁵ (13%) | -97.4 (sd=41.2) | -18% (sd=8) |
| Bowen Broken Bogie | 6.01x10⁵ (10%) | -99.6 (sd=49.8) | -15% (sd=7) |
| Lower Burdekin | 2.03x10⁵ (3%) | -95.4 (sd=49.5) | -15% (sd=8) |
| Entire BDT region | 59.83x10⁵ (100%) | -97.5 (sd=43.9) | -17% (sd=7) |





**Table 6.** Negative trends in area, inter-annual rate, and severity of LNS for river basins of the Burdekin Dry Tropics. sd – standard deviation.

| River basin (in decreasing order of area) | Total area in km$^2$ and percentage of those areas with negative trends | Average trend in gC m$^{-2}$ year$^{-1}$ | Average LNS in gC m$^{-2}$ year$^{-1}$ and as a percentage of reference NPP |
|---|---|---|---|
| Upper Burdekin | 1.26x10$^5$ (24%) | -7.3 (sd = 2.8) | -102.0 (-16%) |
| Belyando | 2.10x10$^5$ (40%) | -8.4 (sd = 3.3) | -134.0 (-27%) |
| Cape Campaspe | 0.71x10$^5$ (14%) | -7.5 (sd = 2.6) | -131.7 (-24%) |
| Suttor | 0.77x10$^5$ (15%) | -13.8 (sd = 10.0) | -184.2 (-34%) |
| Bowen Broken Bogie | 0.22x10$^5$ (4%) | -9.7 (sd = 6.0) | -95.0 (-14%) |
| Lower Burdekin | 0.15x10$^5$ (3%) | -9.5 (sd = 5.6) | -116.0 (-17%) |
| Entire BDT region | 5.21x10$^5$ (100%) | -8.9 (sd = 5.4) | -120.5 (-25%) |





**Table 7.** Positive trends in area, inter-annual rate, and severity of LNS for river basin of the Burdekin Dry Tropics. sd – standard deviation.

| River basin (in decreasing order of area) | Total area in km² and percentage of those areas with positive trends | Average trend in gC m$^{-2}$ year$^{-1}$ | Average LNS in gC m$^{-2}$ year$^{-1}$ and as a percentage of reference NPP |
|---|---|---|---|
| Upper Burdekin | $2.70 \times 10^5$ (27%) | 7.6 (sd = 2.7) | -124.5 (-20%) |
| Belyando | $2.50 \times 10^5$ (25%) | 6.8 (sd = 3.3) | -151.1 (-31%) |
| Cape Campaspe | $1.53 \times 10^5$ (15%) | 7.3 (sd = 3.0) | -113.4 (-21%) |
| Suttor | $1.67 \times 10^5$ (16%) | 8.7 (sd = 3.7) | -139.6 (-26%) |
| Bowen Broken Bogie | $1.46 \times 10^5$ (14%) | 10.1 (sd = 3.6) | -118.4 (-19%) |
| Lower Burdekin | $0.31 \times 10^5$ (3%) | 8.6 (sd = 3.4) | -117.2 (-18%) |
| Entire BDT region | $10.16 \times 10^5$ (100%) | 7.9 (sd = 3.4) | -130.7 (-23%) |





**Table 8.** VAST class comparison with inter-annual trends in LNS and average LNS. sd – standard deviation.

| VAST classes | Average trend in gC m$^{-2}$ year$^{-1}$ | Average LNS in gC m$^{-2}$ year$^{-1}$ | Average LNS as a percentage of reference NPP |
|---|---|---|---|
| 0-'Residual' | 0.3 (sd = 4.7) | -110.2 (sd = 63.7) | -19.7% (sd = 11.1) |
| 1-'Modified' | 1.0 (sd = 4.8) | -110.2 (sd = 61.5) | -19.4% (sd = 10.5) |
| 2-'Transformed' | 1.1 (sd = 5.1) | -115.2 (sd = 62.6) | -21.6% (sd = 11.7) |
| 3-'Replaced' | 0.6 (sd = 6.1) | -123.6 (sd = 66.1) | -24.9% (sd = 12.6) |
| 4-'Removed' | 1.5 (sd = 5.3) | -171.5 (sd = 98.2) | -32.7% (sd = 17.7) |
| 5-'Bare' | -0.9 (sd = 7.3) | -130.2 (sd = 78.6) | -23.9% (sd = 14.5) |



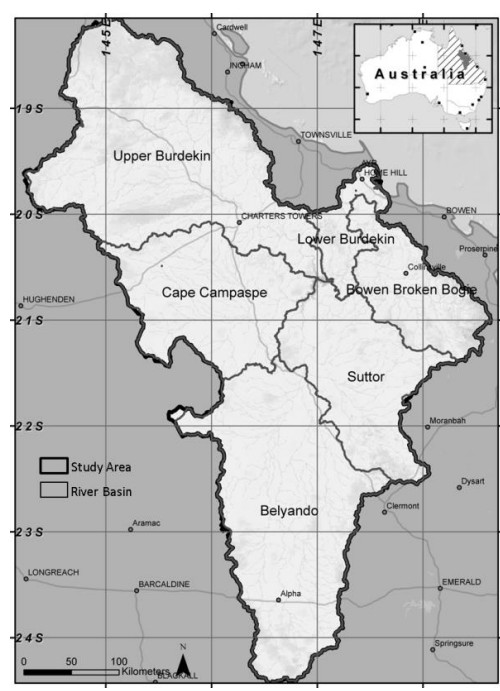

Figure 1.   Location of the Burdekin Dry Tropics (BDT) region in the State of Queensland, Australia, the six major river basins, and major roads and towns.



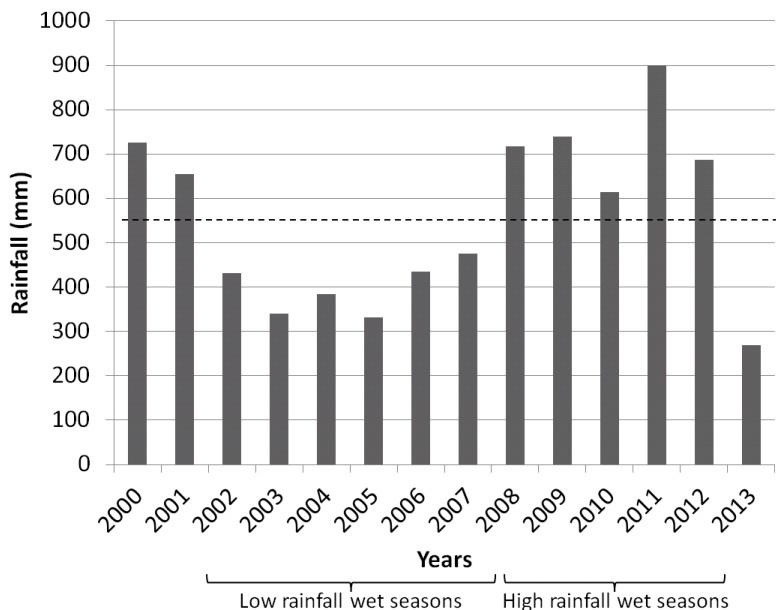

Figure 2. Annual average rainfall in BDT for 2000-2013. The dashed line is the 14 year average.





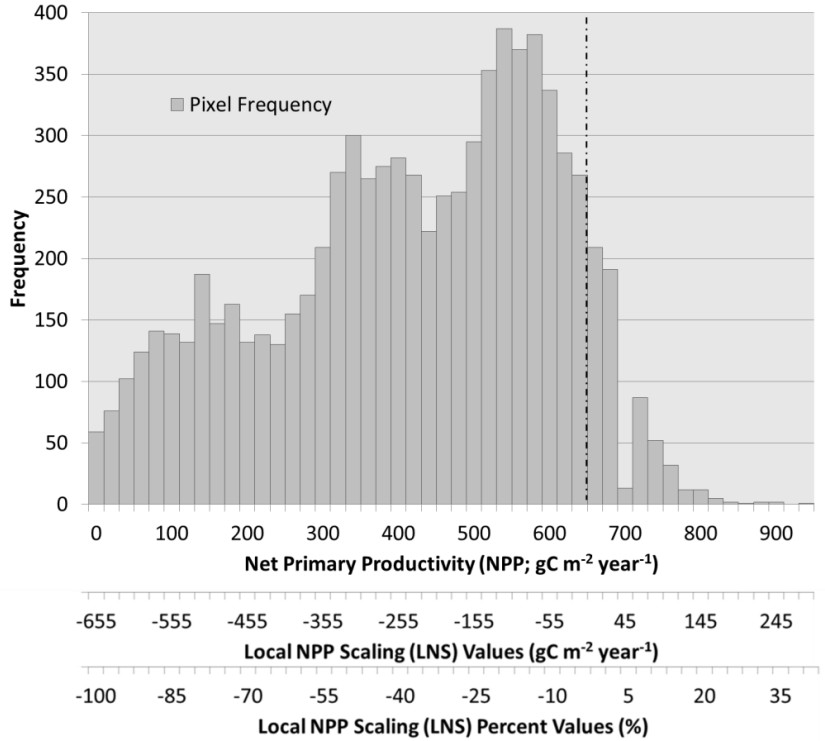

Figure 3. Example of the use of the frequency distribution of NPP of pixels in a single Land
Capability Class (LCC) to calculate Local NPP Scaling (LNS) values. The vertical line
denotes the reference NPP at the 85 percentile of the distribution. The abscissa is
labelled in LNS, NPP and percentage LNS units.





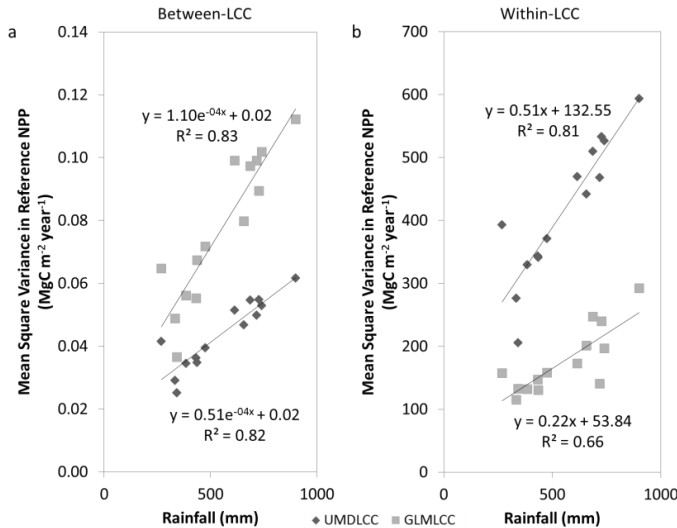

Figure 4. Mean square variance in reference NPP (MgC m$^{-2}$ year$^{-1}$) for UMDLCC and GLMLCC
in relation to rainfall; (a) 'between-LCC' and (b) 'within-LCC' with best-fit regression
lines.



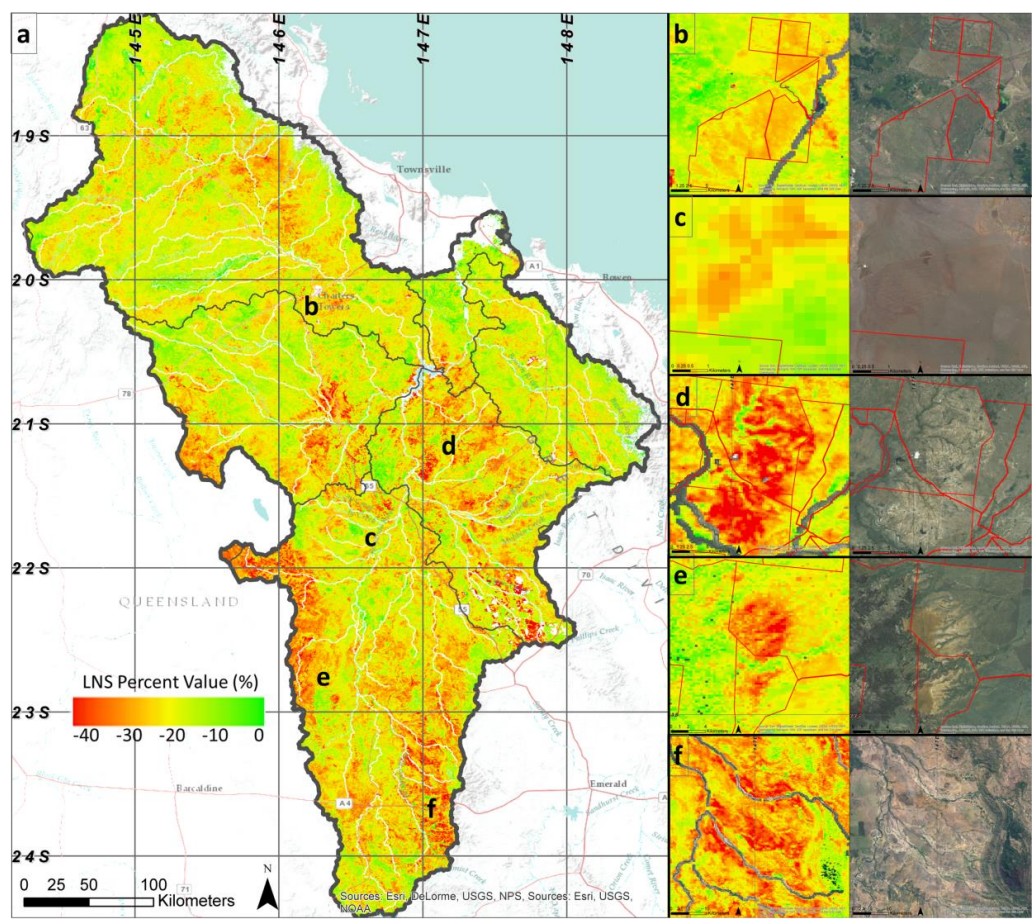

Figure 5. Local Net Production Scaling (LNS) in the Burdekin Dry Tropics (BDT) (a) and enlargements of the areas indicated in (a): (b) high and low LNS values on either side of a station boundary; (c) variation within a single station showing gradients from low to high; (d) low LNS in eroded drainage area; (e) hillslope erosion resulting in bare surface with little to no vegetation cover; (f) area of tree removal with visible erosion and reduced cover. Black lines are the boundaries of river basins and red lines are station boundaries.





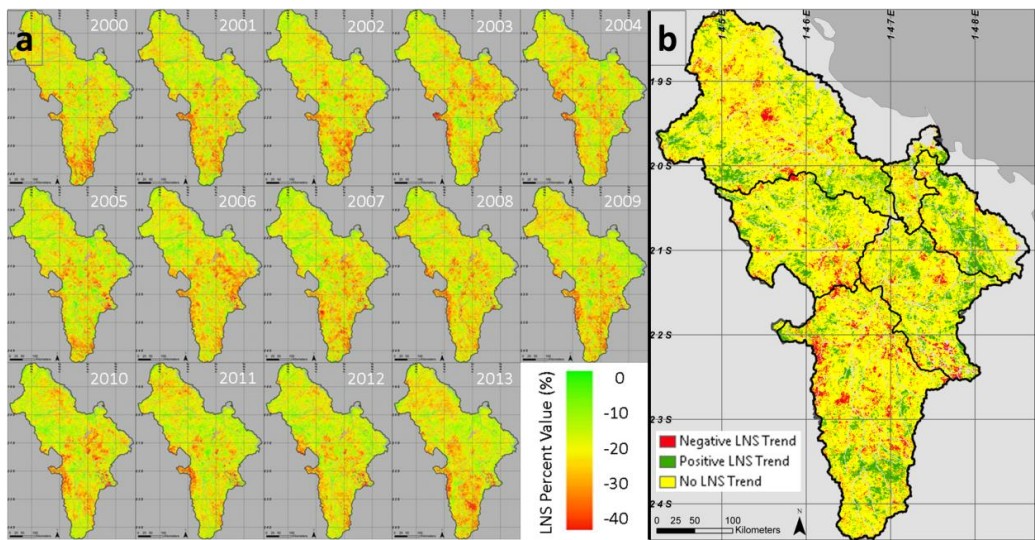

Figure 3. Time-series of maps of the Burdekin Dry Tropics from 2000-2013 showing (a) annual LNS percent values from 2000 to 2013 and (b) inter-annual trends in LNS classified into negative, positive, and no LNS trend.





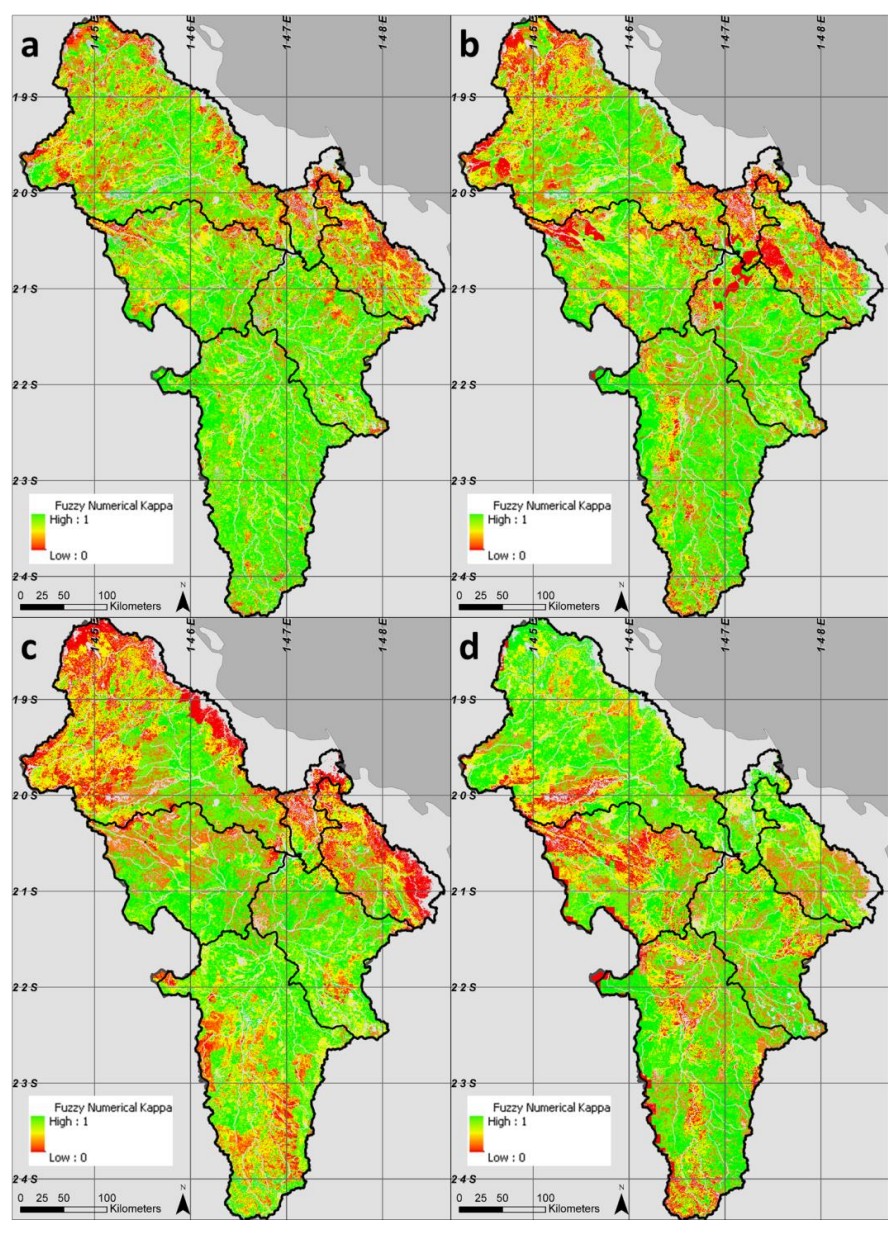

Figure 4. Similarities between (a) annual hillslope erosion, (b) gully density, (c) rainfall
erosivity, and (d) sediment load and percentage LNS as indicated by fuzzy numeric
kappa.

