# Peer review of "Degradation of net primary production in a semi-arid rangeland"

_Biogeosciences, 2015_

## Referee Comment (RC1) · B. K. Wylie (Referee) · 8 Feb 2016

**GENERAL QUALITY**

I liked the approach used in this manuscript but wondered why they did not include slope / aspect criteria in their cluster inputs (big effects in the wester US in arid, boreal, and arctic systems which are ecologically known and accepted. I would be surprised if this was not also true of Australian rangelands). I particularly liked the diligence and innovative summarization of results (pg 8). I found that the Acronyms were numerous and wondered if this journal did a list of acronym section that would have aided the reader? Two prominent things stick out to me: 1) lack of familiarity with the literature and 2) major miss-numbering issues with Figures (assumedly) 6 and 7 (pg s 34 & 35).

**SCIENTIFIC QUESTIONS**
[Figure]

"...there is currently no other method available, LNS was used" pg 3 ln 6. 1) Literature: Prominent articles that I am aware of focused on isolating management are included below. I was surprised the authors only found one or 2 of these. They do cite Wessels et al 2007 & 2008 (pg 2) which seem to me to be a viable and comparable approach):

a. Western US Rangelands i. Wylie, B.K., Boyte, S.P., and Major, D.J., 2012, Ecosystem performance monitoring of rangelands by integrating modeling and remote sensing: Rangeland Ecology and Management, v. 65, no. 4, p. 241-252, at http://dx.doi.org/10.2111/REM-D-11-00058.1. ii. Boyte, S.P., Wylie, B.K., and Major, D.J., 2015, Mapping and monitoring cheatgrass dieoff in rangelands of the Northern Great Basin, USA: Rangeland Ecology and Management, v. 68, no. 1, p. 18-28, at http://dx.doi.org/10.1016/j.rama.2014.12.005. iii. Rigge, M.B., Wylie, B.K., Zhang, L., and Boyte, S.P., 2013, Influence of management and precipitation on carbon fluxes in Great Plains grasslands: Ecological Indicators, v. 34, p. 590-599, at http://dx.doi.org/10.1016/j.ecolind.2013.06.028. iv. Gu, Y.; Wylie, B.K. Detecting ecosystem performance anomalies for land management in the upper Colorado River basin using satellite observations, climate data, and ecosystem models. Remote Sens. 2010, 2, 1880–1891. v. Rigge, M.B., Wylie, B.K., Gu, Y., Belnap, J., Phuyal, K.P., and Tieszen, L.L., 2013, Monitoring the status of forests and rangelands in the western United States using ecosystem performance anomalies: International Journal of Remote Sensing, v. 34, no. 11, p. 4049-4068, at http://dx.doi.org/10.1080/01431161.2013.772311.

b. Boreal forests i. Wylie, B.K., Rigge, M.B., Brisco, B., Murnaghan, K., Rover, J.A., and Long, J.B., 2014, Effects of disturbance and climate change on ecosystem performance in the Yukon River Basin boreal forest: Remote Sensing, v. 6, no. 10, p. 9145-9169, at http://dx.doi.org/10.3390/rs6109145. ii. Wylie, B.K., Zhang, L., Bliss, N.B., Ji, L., Tieszen, L.L., and Jolly, W.M., 2008, Integrating modelling and remote sensing to identify ecosystem performance anomalies in the boreal forest, Yukon River Basin, Alaska: International Journal of Digital Earth, v. 1, no. 2, p. 196-220, at

http://dx.doi.org/10.1080/17538940802038366. iii. NDVI prediction 1. Bunn, A.G., Goetz, S.J. and Fisk J., 2005. Observed and predicted responses of plant growth to climate across Canada. Geophysical Research Letters, 32, L16710, 14.

c. Africa i. Hermann, S.M., Anyamba, A. and Tucker, C.J., 2005. Recent trends in vegetation dynamics in the African Sahel and their relationship to climate. Global Change Biology, 15, 394404. ii. Wessels, K.J., S.D. Prince, et al., 2007, Can human-induced land degradation be distinguished from the effects of rainfall variability? A case study in South Africa. Journal of Arid Environments, 68, 271297. iii. Archer, E.R.M. Beyond the "climate versus grazing" impasse: Using remote sensing to investigate the effects of grazing system choice on vegetation cover in the eastern Karoo. J. Arid Environ. 2004, 57, 381–408.

Pg 4 ln 8-9: It seems that the nearest neighbor approach would merely retain the blockiness of the 5 k x 5 k data. Why not use an interpolation to smooth 5k 5k pixel boundaries? Say cubic or bilinear interpolation? Why not include slope and aspect? Known ecological difference occur related to certain conditions (south vs north aspect with moderate to steep slopes) in many ecosystems, particularly temperature limited (Arctic and Boreal) and moisture limited ones. In the northern hemisphere you would be showing all southern aspects as degraded when they are just drier because of higher transpiration demands from higher temperatures than north facing slopes. The same would be true for southern hemisphere, only with north slopes being drier..

Pg 7 ln 15: it would be interesting to field check these all year reference sites.

Pg 11 ln 1: Convection thunderstorm precipitation is HARD to map accurately. Often in remote areas with few weather stations, gridded precipitation can be unreliable when distant from a weather station.

Pg 11 ln 17: "largest spatial variations" Think of ecological tendencies for larger means to have larger variances. What if you use CV (coefficient of variation)?
[Figure]

Pg 11 ln22-27: "∼ need for comparison to pixel based estimated productivity" This sounds exactly what Wylie et al, Rigge et al. Gu et al. are doing but instead of a process-based model (classically heavily depend on precipitation which is notoriously problematic to map in remote landscapes) data driven regression trees were used to predict undisturbed productivity or potential productivity.

Pg 12 ln 39: "∼relationship to hillslope erosion)" Not convinced unless slope/aspect are taken into account in LCC.

TECHNICAL CORRECTIONS

The miss-numbered figures 6 and 7 seemed out of place in an otherwise very thoughtful paper.

Pg4 ln 36: Why not see if the 2 difference clusters/land groupings are consistent spatially? "mean square variance of their maximum NPP" was confusing. Re-word? I was confused if you only had one max value per LCC how you could get a variance of, that but later it became clear that you were looking a the variance of max-each pixel in the LCC. One statistical buddy told me that maximized variables have weird statistical properties and should be avoided (you also mention the maximum is susceptible to selecting "outliers"). We have used mean values from the upper quartile to avoid such issues. I see later (Fig3) you use 85 percentile. Why did you choose to use the maximum for the difference in clusters vs land grouping? I think it is "OK" but if you apply this elsewhere I would consider changing this.

Pg 5 ln 27-28: Why not downscale 250 m to 1km, run the regression at 1km (ndvi vs npp)? At least then you are comparing apples to apples... 250m variation is going to just be different than 1 km variation.

Pg 6 ln 4: "reference pixels" Glad to see acknowledgement of the limitations but I do not think the readers understand where the reference pixels come from because Fig 3 has not been presented. I was confused at this point before Fig 3 was introduced.

[Figure]

(also true at Pg 4 ln 12)

Pg 7 ln 1-7: In the US, the BLM (major federal land management agency for western arid rangelands) has locked in as percent bare ground as a good indicator of range condition. Are there any estimates of this you could use? I know there is a soil property mapping effort/research going on in Australia (Henderson et al. 2005, Geoderma 124:383-398) or continuous land cover (http://landcover.usgs.gov/pdf/canopy_density.pdf; http://glcf.umd.edu/data/treecover/) which could be used? Maybe remote sensing vegetation indices??. I am concerned that by not including slope and aspect in you r LCC determination that you maybe incorrectly identifying drier north slopes as degraded.. I guess your soil erodibility data is OK but soil texture differences could be a major driver in those determinations, not management. . ..

Pg 7 ln 24: "but between-LCC" Fig 4 miss labeled or text is wrong. Fig 4b has these statistics but was labeled "within LCC". I think the association with rain does not add much, particularly to assess the 2 clustering approaches. Why not plot variance vs your maximum NPP or reference NPP or mean cluster NPP? I think you are just using precipitation as proxy for productivity here. Higher variances with higher means is a common phenomenon in ecological data, thus often the coefficient of variance is used.

Pg 9 ln 16. I like your quantification of degradation in units of NPP.

Pg 9 ln 10: Fig 5f: I think I see possible difference associated with slope / aspect differences. . .

---

## Referee Comment (RC2) · Anonymous Referee #2 · 6 Apr 2016

General comments

Overall I found that the manuscript accomplished its stated objectives using a novel approach to address the main limitation of LNS, was for the most part clearly written, and stands to make a contribution both conceptually in understanding the prevalence and rates of degradation, as well as methodologically through improving remotely sensed rangeland monitoring, areas of research in much need of advancement. In order of importance, I particularly welcome the use of shifting annual reference NPP pixels to demonstrably improve LCC classification (although the reliability of some reference sites might be questioned), the attempt to evaluate LCC classification using independently-derived datasets measuring elements of land potential, and the generally pragmatic, conservative decisions made at several steps that improve the robustness of the analysis. This being said, I do think the manuscript could be stronger in

several respects. Some assumptions are unaddressed or under-stated, the precipitation gradient in the region was not well utilized, and the organization and presentation of results could be much clearer, especially the tables.

Specific comments

"The method is limited spatially only by the capacity to classify the land," (page 1, line 24): I'm not sure exactly what this means, but I doubt it's true. A key assumption of the analysis is the accuracy of MODIS NPP in the study area. In tropical grasslands both dry and wet, this data can be unreliable for different reasons. In fact, it could explain why weak NPP and degradation gradients were observed. If there are relevant assessments for the region, cite them. If not, best to evaluate the MODIS data to the extent feasible, or use more than one method for NPP.

Another assumption is that use of foliage projective cover (FPC) in defining LCCs did not unduly alter the analysis and conclusions. The soil and weather data are arguably independent of degradation, vegetation condition is not. While I understand the logic in using FPC, and it is not necessarily problematic, I'd prefer a mention of what factors the classification was robust to when included (or not), and a correlation matrix of factors used for LCC classification at minimum.

The manuscript missed an opportunity to use the (large) rainfall gradient in the region productively. Analyses were presented and interpreted at river basin scales, which to me is not the natural unit of aggregation for analysis in this case (as hydrology is not the primary focus). I would have preferred to see, for example, mean precipitation isohyets delineated at increments from the coast, and degradation trends analyzed specifically within and between these areas. Addressing rainfall explicitly would have greatly increased the amount of information produced by the analysis.

With regard to the manuscript's presentation, most importantly, some numbers do not appear to add up, and their derivation must be checked and clarified. Table 2 gives -1.71 (non-degraded) and -3.90 (degraded) MgCm-2yr-1 as the average LNS values

for these 2 degradation classes, which firstly form the basis for the whopping "2.14 MgCm-2yr-1" typo (hopefully) in the abstract, text, and Table 2. Secondly, Tables 5 and 4 respectively provide -97.5 (non-degraded) -209.1 (degraded) gCm-2yr-1 as apparently the same values. If river basins must be used to organize the tables, they would be more effective if reorganized. Cutting down the table text and combining tables to align figures on degraded area, trend categories, and/or degradation classes would present the results much more clearly. Finally, including the reference NPP, rainfall, or some other indicator of overall productivity potential would make the reported values more meaningful. Alternatively, summarize such relevant statistics by basin in an appendix. Finally, it would have been nice to see a map with degradation class-by-trend combinations, to show where is degraded, where is being degraded, and where is recovering. Finally, some tables and figures should be shifted to supplementary materials.

Technical corrections

Page 4, line 27: GLMLCC is static, not dynamic as in the UMDLCC approach here Page 5, line 18: "soil erodibility" was apparently not used Page 5, line 34: missing end parenthesis; what is a "distributary"? Page 7, line 5: "accounts," not "allows" Page 8, lines 3-4: as compared to a reference mean of . . . what? Page 8, lines 9-10: reword; typos Page 8, lines 14-16: "had"? Page 8, line 21: "smaller"? I think you mean "lower" Page 9, lines 4-10: Does not match the figure legend. Page 10, line 11: "were occurred in"? Page 10, line 18-24: naturally 'bare' ground is undergoing degradation? Page 10, line 33-36: reword Page 12, line 12: Table 2, not Table 1 Page 12, lines 4-20: These numbers do not match the tables. Also, permanent degradation cannot be inferred here. Page 12, line 29: "strong" correlation? What is the evidence? Page 32: Clarify that points are years, not LCCs or something else Page 34: Figure 3. . .? Page 35: Figure 4. . .?

---

## Author Response (AR1)

**Note**

The authors are most grateful for these unusually detailed and comprehensive reviews.

··············································································

Referees comments are in plain text, authors' responses are in bold below the referee
comment.

**Referee #1**

B.K. Wylie

Scientific Questions

". . .there is currently no other method available, LNS was used" pg 3 ln 6. 1) Literature:
Prominent articles that I am aware of focused on isolating management are included below.
I was surprised the authors only found one or 2 of these. They do cite Wessels et al 2007 &
2008 (pg 2) which seem to me to be a viable and comparable approach):

a. Western US Rangelands

i. Wylie, B.K., Boyte, S.P., and Major, D.J., 2012, Ecosystem performance monitoring of
rangelands by integrating modeling and remote sensing: Rangeland Ecology and
Management, v. 65, no. 4, p. 241-252, at http://dx.doi.org/10.2111/REM-D-11-00058.1.

ii. Boyte, S.P., Wylie, B.K., and Major, D.J., 2015, Mapping and monitoring cheatgrass dieoff in
rangelands of the Northern Great Basin, USA: Rangeland Ecology and Management, v. 68, no.
1, p. 18-28, at http://dx.doi.org/10.1016/j.rama.2014.12.005.

iii. Rigge, M.B., Wylie, B.K., Zhang, L., and Boyte, S.P., 2013, Influence of management and
precipitation on carbon fluxes in Great Plains grasslands: Ecological Indicators, v. 34, p. 590-
599, at http://dx.doi.org/10.1016/j.ecolind.2013.06.028.

iv. Gu, Y.; Wylie, B.K. Detecting ecosystem performance anomalies for land management in
the upper Colorado River basin using satellite observations, climate data, and ecosystem
models. Remote Sens. 2010, 2, 1880–1891. v. Rigge, M.B., Wylie, B.K., Gu, Y., Belnap, J.,
Phuyal, K.P., and Tieszen, L.L., 2013, Monitoring the status of forests and rangelands in the
western United States using ecosystem performance anomalies: International Journal of
Remote Sensing, v. 34, no. 11, p. 4049-4068, at
http://dx.doi.org/10.1080/01431161.2013.772311.

b. Boreal forests

i. Wylie, B.K., Rigge, M.B., Brisco, B., Murnaghan, K., Rover, J.A., and Long, J.B., 2014, Effects of disturbance and climate change on ecosystem performance in the Yukon River Basin boreal forest: Remote Sensing, v. 6, no. 10, p. 9145-9169, at http://dx.doi.org/10.3390/rs6109145.

ii. Wylie, B.K., Zhang, L., Bliss, N.B., Ji, L., Tieszen, L.L., and Jolly, W.M., 2008, Integrating modelling and remote sensing to identify ecosystem performance anomalies in the boreal forest, Yukon River Basin, Alaska: International Journal of Digital Earth, v. 1, no. 2, p. 196-220, at http://dx.doi.org/10.1080/17538940802038366.

iii. NDVI prediction 1. Bunn, A.G., Goetz, S.J. and Fisk J., 2005. Observed and predicted responses of plant growth to climate across Canada. Geophysical Research Letters, 32, L16710, 14.

c. Africa

i. Hermann, S.M., Anyamba, A. and Tucker, C.J., 2005. Recent trends in vegetation dynamics in the African Sahel and their relationship to climate. Global Change Biology, 15, 394404.

ii. Wessels, K.J., S.D. Prince, et al., 2007, Can humaninduced land degradation be distinguished from the effects of rainfall variability? A case study in South Africa. Journal of Arid Environments, 68, 271297. iii. Archer, E.R.M. Beyond the "climate versus grazing" impasse: Using remote sensing to investigate the effects of grazing system choice on vegetation cover in the eastern Karoo. J. Arid Environ. 2004, 57, 381–408.

**Authors' Response: The referee lists very relevant studies that do seek to isolate management effects from climatic variability. The referee also correctly points out that only two of these works were cited in the manuscript. The two that were cited are the most closely aligned with the focus of the current manuscript. However, we believe that the reader missed the point made was in regard to the limitations which exist in current methods. For example, in the first paper provided, a rule-based approach was used. The aim of the current manuscript was to produce a repeatable method which does not rely of intimate knowledge of the rangeland system. To accomplish this we sought to allow objective, unsupervised data clustering to decide homogeneous units. Furthermore, the current manuscript develops land capability classes which are not a reflection on vegetation types whatsoever, as had been presented in many of the supplied references, but rather is solely based upon measurable characteristics of the regional environment. This way long term transitions in land condition which result in changes in vegetation type (e.g. invasive species and encroachment of unpalatable woody species) are included in our definition of degradation.**

Pg 4 ln 8-9: It seems that the nearest neighbor approach would merely retain the blockiness of the 5 k x 5 k data. Why not use an interpolation to smooth 5k 5k pixel boundaries? Say cubic or bilinear interpolation? Why not include slope and aspect? Known ecological difference occur related to certain conditions (south vs north aspect with moderate to steep slopes) in many ecosystems, particularly temperature limited (Arctic and Boreal) and

moisture limited ones. In the northern hemisphere you would be showing all southern aspects as degraded when they are just drier because of higher transpiration demands from higher temperatures than north facing slopes. The same would be true for southern hemisphere, only with north slopes being drier..

5 **Authors' Response: Interpolation was not used because the data was used in a cluster algorithm, which sought to distinguish homogeneous areas based upon actual data. In the Burdekin, as in the rest of Australia, a high quality climate data set exists for which necessary data smooth already exist. The use the average mean LNS value over multiple years also creates a smoothing effect that is more closely related to actual**

10 **climate values.**

**Authors' Response: The study region is largely flat so the use of slope and aspect only drive up model error. Furthermore soil properties used to define land capability classes are related to topographic features. Additionally measure of soil erosion, another closely related variable with slope and aspect were used for comparison with**

15 **LNS results.**

**Authors' Response: Finally, fine scale differences in aspect and slope are a naturally occurring phenomenon in each LCC. Low LNS values in these areas are also a valuable indication of degradation and may be compared across LCCs.**

Pg 7 ln 15: it would be interesting to field check these all year reference sites.

20 **Authors' Response: Great point, noted.**

Pg 11 ln 1: Convection thunderstorm precipitation is HARD to map accurately. Often in remote areas with few weather stations, gridded precipitation can be unreliable when distant from a weather station.

**Authors' Response: The Australian climate data has an overall accuracy of 84% and**

25 **the study region falls in an area where a dense network of weather stations exist.**

Pg 11 ln 17: "largest spatial variations" Think of ecological tendencies for larger means to have larger variances. What if you use CV (coefficient of variation)?

**Authors' Response: This comment is in response to the standard deviation values in northern basins and the proposed CV provides the same information, specifically**

30 **because CV is simply the standard deviation divided by the mean.**

Pg 11 ln22-27: "~ need for comparison to pixel based estimated productivity" This sounds exactly what Wylie et al, Rigge et al. Gu et al. are doing but instead of a process-based model (classically heavily depend on precipitation which is notoriously problematic to map in remote landscapes) data driven regression trees were used to predict undisturbed

35 productivity or potential productivity.

**Authors' Response: A data driven regression tree is another future alternative to the development of LCCs.**

Pg 12 ln 39: "~relationship to hillslope erosion)" Not convinced unless slope/aspect are taken into account in LCC.

**Authors' Response: In figure 7, the agreement is presented. As stated earlier elements related to topography may be related to the chosen soil properties.**

TECHNICAL CORRECTIONS

The miss-numbered figures 6 and 7 seemed out of place in an otherwise very thoughtful paper.

**Authors' Response: Corrected**

Pg4 ln 36: Why not see if the 2 difference clusters/land groupings are consistent spatially? "mean square variance of their maximum NPP" was confusing. Re-word? I was confused if you only had one max value per LCC how you could get a variance of, that but later it became clear that you were looking a the variance of max-each pixel in the LCC. One statistical buddy told me that maximized variables have weird statistical properties and should be avoided (you also mention the maximum is susceptible to selecting "outliers"). We have used mean values from the upper quartile to avoid such issues. I see later (Fig3) you use 85 percentile. Why did you choose to use the maximum for the difference in clusters vs land grouping? I think it is "OK" but if you apply this elsewhere I would consider changing this.

**Authors' Response: The maximum referred to in the text is the best estimator of the potential value, which is the 85 percentile. NPP values higher than this were omitted (as stated in the manuscript), so no assumptions are made about their distribution. The goal was to 'model' the unmanaged portion of each LCC. The mean square variation was used for exactly the reason the referee pointed out (i.e. minimizing the effect of outliers while still analyzing variation within the population of maximum values). The differences in the maximum values found were then assumed to be naturally occurring differences, unrelated to management. In a highly managed rangeland such as the BDT, this assumption should hold true.**

Pg 5 ln 27-28: Why not downscale 250 m to 1km, run the regression at 1km (ndvi vs npp)? At least then you are comparing apples to apples... 250m variation is going to just be different than 1 km variation.

**Authors' Response: This was done, the regression was performed at 1km, then downscaled to 250m. The spatial scaled of 250m was used because degradation related human management is most relevant at spatial scales finer than 1km for many reasons (e.g. grazing enclosure size, differences across property boundaries, highly variable vegetation, etc.)**

Pg 6 ln 4: "reference pixels" Glad to see acknowledgement of the limitations but I do not think the readers understand where the reference pixels come from because Fig 3 has not been presented. I was confused at this point before Fig 3 was introduced. (also true at Pg 4 ln 12)

**Authors' Response: Pg 6 ln 4: Changed "reference pixels" to "the potential"**

**Authors' Response: Pg 6 ln 12: Figure 3 is introduced on the same line at Pg 4 ln 12, so no correction is needed there**

Pg 7 ln 1-7: In the US, the BLM (major federal land management agency for western arid rangelands) has locked in as percent bare ground as a good indicator of range condition. Are there any estimates of this you could use? I know there is a soil property mapping effort/research going on in Australia (Henderson et al. 2005, Geoderma 124:383-398) or continuous land cover (http://landcover.usgs.gov/pdf/canopy_density.pdf; http://glcf.umd.edu/data/treecover/) which could be used? Maybe remote sensing vegetation indices??. I am concerned that by not including slope and aspect in you r LCC determination that you maybe incorrectly identifying drier north slopes as degraded.. I guess your soil erodibility data is OK but soil texture differences could be a major driver in those determinations, not management. . ..

**Authors' Response: Bare ground data in Australia is typically limited to scales that aren't relevant to regional degradation mapping (e.g. 30m - Landsat, 20m – ASTER versus the 250m –MODIS used in the manuscript). The major problem with using additional aspects of the degradation, such as bare ground is that, if they may be derivatives of the same vegetation indices used for validation of the argument is circular!.**

**Authors' Response: The substantial agreement of hillslope erosion, a metric highly related to slope, should alleviate most of the danger. To a lesser extent it is impossible to remove all elements of weather.**

Pg 7 ln 24: "but between-LCC" Fig 4 miss labeled or text is wrong. Fig 4b has these statistics but was labeled "within LCC".

**Authors' Response: Pg 7 ln 24: Changed "Figure 4a" to "Figure 4b"**

**Authors' Response: Pg 7 ln 24: Changed "Figure 4b" to "Figure 4a"**

I think the association with rain does not add much, particularly to assess the 2 clustering approaches. Why not plot variance vs your maximum NPP or reference NPP or mean cluster NPP? I think you are just using precipitation as proxy for productivity here. Higher variances with higher means is a common phenomenon in ecological data, thus often the coefficient of variance is used.

**Authors' Response: Precipitation was used because it is the primary environmental factor which drives differences in potential productivity. This means that if the LCCs**

**can reduce the within-group variation and maximize the between group variation, they are outperforming the GLM map. This gets to a previous point made in the manuscript that it is impossible for all symptoms of the environment to be removed, instead we must manage the impact of the most important environment variables**

5    Pg 9 ln 16. I like your quantification of degradation in units of NPP.

**Authors' Response: Thanks.**

Pg 9 ln 10: Fig 5f: I think I see possible difference associated with slope / aspect differences. . .

**Authors' Response: The river area was masked so steep slopes associated with**
10   **riparian zone were minimized. As the text states, it was only the interfluves that were included. It is true that severe erosion can take place on river banks and riparian health has become a major problem in the study region and has resulted in abundant resources to remedy resulting erosion from these zones. This type of degradation was excluded owing to its finer scale than the 250m data that were available.**

**Referee # 2**

Anonymous Referee #2

General comments

20   Overall I found that the manuscript accomplished its stated objectives using a novel approach to address the main limitation of LNS, was for the most part clearly written, and stands to make a contribution both conceptually in understanding the prevalence and rates of degradation, as well as methodologically through improving remotely sensed rangeland monitoring, areas of research in much need of advancement. In order of importance, I
25   particularly welcome the use of shifting annual reference NPP pixels to demonstrably improve LCC classification (although the reliability of some reference sites might be questioned), the attempt to evaluate LCC classification using independently-derived datasets measuring elements of land potential, and the generally pragmatic, conservative decisions made at several steps that improve the robustness of the analysis. This being said,
30   I do think the manuscript could be stronger in several respects. Some assumptions are unaddressed or under-stated, the precipitation gradient in the region was not well utilized, and the organization and presentation of results could be much clearer, especially the tables.

Specific comments

"The method is limited spatially only by the capacity to classify the land," (page 1, line 24): I'm not sure exactly what this means, but I doubt it's true. A key assumption of the analysis is the accuracy of MODIS NPP in the study area. In tropical grasslands both dry and wet, this data can be unreliable for different reasons. In fact, it could explain why weak NPP and

5 degradation gradients were observed. If there are relevant assessments for the region, cite them. If not, best to evaluate the MODIS data to the extent feasible, or use more than one method for NPP.

**Authors' Response: Regarding the spatial limitation of LNS, the limitation is the spatial resolution of the satellite data that are used. If Landsat data were frequent**
10 **enough to be used to estimate NPP, and was available with adequate frequency, the LNS analysis could be undertaken at that scale.**

**Authors' Response: The referee also states a key difficulty for virtually all remote sensing studies - reliability of the data. While the errors in the LNS procedure far outweigh those associated with the sensor, it is nevertheless true that the MODIS NPP**
15 **product, based as it is on a light-use efficiency model, is frequently inaccurate. In the present case it was assumed that NPP errors would be minimized in the limited area (compared with global) that were analyzed.**

**Authors' Response: We do agree that rewording/removing will help avoid additional confusion.**

20 **Authors' Response: Pg 1 line 24: Sentence deleted. "The method is limited spatially only by the capacity to classify the land."**

Another assumption is that use of foliage projective cover (FPC) in defining LCCs did not unduly alter the analysis and conclusions. The soil and weather data are arguably independent of degradation, vegetation condition is not. While I understand the logic in
25 using FPC, and it is not necessarily problematic, I'd prefer a mention of what factors the classification was robust to when included (or not), and a correlation matrix of factors used for LCC classification at minimum.

**Authors' Response: Foliage projective cover was used as a reference point to start to separate pre-2000 vegetation groupings. The point was to limit the opportunity of**
30 **different, existing vegetation groups from being compared with each other and thus minimize false interpretation as degradation.**

**Authors' Response: A correlation matrix for 50 classes for each year over 14 years would be tedious (50x14= 700 cells) for the reader to evaluate. A correlation matrix for just one LCC could be included but would not be representative of any other LCC.**

35 The manuscript missed an opportunity to use the (large) rainfall gradient in the region productively. Analyses were presented and interpreted at river basin scales, which to me is not the natural unit of aggregation for analysis in this case (as hydrology is not the primary focus). I would have preferred to see, for example, mean precipitation isohyets delineated at

increments from the coast, and degradation trends analyzed specifically within and between these areas. Addressing rainfall explicitly would have greatly increased the amount of information produced by the analysis.

**Authors' Response: Climate (including rainfall) was included in the creation of LCCs, although not in the form of climatological isohyets across the entire region, rather as annual rainfall. It is true, however, that long-term environmental differences, as captured to some extent by climatology, may create more homogeneity within LCCs, and we acknowledge that this should be explored in future studies.**

**Authors' Response: Second, river basins provided a more natural comparison with management units which are of interest to policy-makers as well as managers and an important factor of concern in the Burdekin Dry Tropics is erosion leading to sediment transport, as mentioned in the Introduction, which contributes to the silting of the Great Barrier Reef.**

With regard to the manuscript's presentation, most importantly, some numbers do not appear to add up, and their derivation must be checked and clarified. Table 2 gives -1.71 (non-degraded) and -3.90 (degraded) MgCm-2yr-1 as the average LNS values for these 2 degradation classes, which firstly form the basis for the whopping "2.14 MgCm-2yr-1" typo (hopefully) in the abstract, text, and Table 2.

**Authors' Response: The 2.08 MgCm-2yr-1 is the average value for the entire study region, not the total. We think this is clear in the table.**

Secondly, Tables 5 and 4 respectively provide -97.5 (non-degraded) -209.1 (degraded) gCm-2yr-1 as apparently the same values. If river basins must be used to organize the tables, they would be more effective if reorganized. Cutting down the table text and combining tables to align figures on degraded area, trend categories, and/or degradation classes would present the results much more clearly. Finally, including the reference NPP, rainfall, or some other indicator of overall productivity potential would make the reported values more meaningful. Alternatively, summarize such relevant statistics by basin in an appendix.

**Authors' Response: The -209.1 gCm-2y-1 value from Table 4 refers to degraded areas, while the -97.5 value from Table 5 refers to the non-degraded areas – as the referee points out - but we are unsure why there could be confusion regarding these. Tables 4 and 5 are straightforward, presenting the average NPP loss, percentage loss and the area affected in each basin and the entire region. The point is that each river basin has different degrees of degradation and that degradation may be interpreted differently (e.g. NPP loss, percent loss) for each basin.**

**Authors' Response: Tables could be combined, but removing key data such as the percent NPP loss would make for confusing analysis because LNS cannot be reliably interpreted across an LCC without using a scaled calculation of loss, such as a**

**percent. Also NPP loss is essential for evaluation because it ties the results to a physical metric which may be compared to other land condition assessments. Part of the new approach presented in the manuscript is the scaled values of NPP and how they are interpreted. The LNS values represent how far the observed NPP is from the reference NPP.**

Finally, it would have been nice to see a map with degradation class-by trend combinations, to show where is degraded, where is being degraded, and where is recovering.

**Authors' Response: These were presented separately to avoid repeating the results.**

Finally, some tables and figures should be shifted to supplementary materials.

**Authors' Response: The tables and figures have been reviewed with this in mind and we concluded they are sufficiently important to the text that they are better left where they are. Their inclusion will not make the paper unusually long.**

Technical corrections

Page 4, line 27: GLMLCC is static, not dynamic as in the UMDLCC approach here

**Authors' Response: Changed this text to make that distinction once again, although it was implied in the Methods and made explicitly in the Discussion.**

Page 5, line 18: "soil erodibility" was apparently not used

**Authors' Response: Soil bulk density, soil water holding capacity and clay percentage were used in the LCCs. Soil erodibility was used (see figure 7) in the evaluation of LNS results.**

Page 5, line 34: missing end parenthesis; what is a "distributary"?

**Authors' Response: Page 5, line 34: Changed "distributary" to "tributary"**

Page 7, line 5: "accounts," not "allows"

**Authors' Response: Page 7, line 5: Changed "allows" to "accounts"**

Page 8, lines 3-4: as compared to a reference mean of . . . what?

**Authors' Response: Sorry, we can't find this text.**

Page 8, lines 9-10: reword; typos

**Authors' Response: Page 8, lines 9-10: Changed "The sum of LNS values for entire class, as opposed to LNS per unit area revealed how the importance the size of each class in contributing to the overall reduction in NPP." to "The sum of LNS values for an entire class, as opposed to the LNS value per unit area, revealed the importance of class size in the overall reduction in NPP."**

Page 8, lines 14-16: "had"?

**Authors' Response: Page 8, lines 14-16: Changed "had" to "were"**

Page 8, line 21: "smaller"? I think you mean "lower"

**Authors' Response: Page 8, line 21: changed "smaller" to "lower"**

5   Page 9, lines 4-10: Does not match the figure legend.

**Authors' Response: Page 9, lines 4-10: Changed "Among degraded areas there was evidence of managed grazing, including abrupt differences in LNS along station boundaries (Figure 5b), but there were also gradients of LNS within some stations (Figure 5c), and others with low LNS spread across boundaries (Figure 5d). Other**
10  **areas with evidence of management included forest clearing (Figure 5e) near station boundaries. There were also locations classified as degraded with little evidence of direct grazing management such as between the drainage lines of streams (Figure 5f)." to "Among degraded areas there was evidence of managed grazing, including abrupt differences in LNS along station boundaries (Figure 5b), but there were also**
15  **gradients of LNS within a single station (Figure 5c), and others with low LNS spread across multiple boundaries (Figure 5d). Other areas with evidence of management included forest clearing (Figure 5e) near station boundaries. There were also locations classified as degraded with little evidence of direct grazing management such as between the drainage lines of streams (Figure 5f)."**

20  Page 10, line 11: "were occurred in"?

**Authors' Response: Page 10, line 11: Changed "were occurred in" to "occurred"**

Page 10, line 18-24: naturally 'bare' ground is undergoing degradation?

**Authors' Response: Page 10, line 18-24:Changed "The only negative trend was in the 'bare' class while 'removed' had the largest positive trend." to "The only negative**
25  **trend was in the 'bare' class, presumably an indication that a small amount of vegetation was present, while 'removed' had the largest positive trend."**

Page 10, line 33-36: reword

**Authors' Response: Page 10, line 33-36: Changed "This indicates that degradation, as detected with LNS, were sites that were persistently below the potential, not simply**
30  **subject to some short-term environmental deficiency, such a single-year with spatially patchy lower rainfall." to "This indicates that degradation, as detected with LNS, corresponded to sites that were persistently below the potential. This emphasized that these sites were not simply subject to some short-term environmental deficiency, such a single-year with spatially patchy lower rainfall."**

35  Page 12, line 12: Table 2, not Table 1

**Authors' Response: Page 12, line 12: Changed "Table 1" to "Table 2"**

Page 12, lines 4-20: These numbers do not match the tables.

**Authors' Response: The numbers do match, but I will ensure the number is presented exactly as in the table**

5 **Authors' Response: Page 12, lines 4-20: Changed "65%" to "65.3"**

Also, permanent degradation cannot be inferred here.

**Authors' Response: It is inferred owing to the irreversible nature of degradation**

**Authors' Response: Page 12, line 15: Changed "presumably" to "a possible indicator"**

Page 12, line 29: "strong" correlation? What is the evidence?

10 **Authors' Response: The evidence is in table 8.**

**Authors' Response: Page 12, line 29: Changed "strong correlation" to "good agreement" to be more precise.**

Page 32: Clarify that points are years, not LCCs or something else

**Authors' Response: Page 32: Changed "...lines." To "...lines for each year 2000 to**
15 **2013."**

Page 34: Figure 3. . .?

**Authors' Response: Previously Corrected**

Page 35: Figure 4. . .?

**Authors' Response: Previously Corrected**

**Degradation of net primary production in a semi-arid rangeland**

25 **H. Jackson[1] and S. D. Prince[1]**
[1]University of Maryland, College Park, Maryland, 20742, U.S.A.

*Correspondence to*: Hasan Jackson (hjackso1@umd.edu)

**Abstract.** Anthropogenic land degradation affects many biogeophysical processes including reductions of net primary production (NPP). Degradation occurs at scales from
5   small fields to continental and global. While measurement and monitoring of NPP in small areas is routine in some studies, for scales larger than 1km$^2$, and certainly global, there is no regular monitoring and certainly no attempt to measure degradation. Quantitative and repeatable techniques to assess the extent of deleterious effects and monitor changes are needed to evaluate its effects on, for example, economic yields of
10   primary products such as crops, lumber and forage, and as a measure of land surface properties which are currently missing from dynamic global vegetation models, assessments of carbon sequestration and land surface models of heat, water, and carbon exchanges. This study employed the Local NPP Scaling (LNS) approach to identify patterns of anthropogenic degradation of NPP in the Burdekin Dry Tropics (BDT) region
15   of Queensland, Australia from 2000 to 2013. The method starts with land classification based on the environmental factors presumed to control (NPP) to group pixels having similar potential NPP. Then, satellite remotely sensing data were used to compare actual NPP with its potential. The difference in units of mass of carbon and percentage loss wereas the measure of degradation.
20    The entire BDT (7.45x10$^6$ km$^2$) was investigated at a spatial resolution of 250x250m. The average annual reduction in NPP due to anthropogenic land degradation in the entire BDT was -2.14 MgC m$^{-2}$ year$^{-1}$ or 17% of the non-degraded potential, and the total reduction was -214 MgC year$^{-1}$. Extreme average annual losses of 524.8 gC m$^{-2}$ year$^{-1}$ were detected. Approximately 20% of the BDT was classified as
25   'degraded'. Varying severities and rates of degradation were found among the river basins, of which the Belyando and Suttor were highest. Inter-annual, negative trends in reductions of NPP, occurred in 7% of the entire region, indicating on-going degradation. There was evidence of areas that were in a permanently degraded condition. The findings provide strong evidence and quantitative data for reductions in NPP related to
30   anthropogenic land degradation in the BDT.

**Commented [HJ1]:** The referee presents a key difficulty for virtually all remote sensing studies: reliability of remotely sensed data. There is unreliability associated with MODIS NPP which is reported in hhhhhh. Spatially unreliability is difficult to quantify due to the many reason it can occur. It should be pointed out that MODIS NPP data has been used in many remote sensing applications in relevant journal articles, including those cited in the manuscript Wessels 2008, Prince et al. 2009.
I do agree that rewording/removing will help avoid additional confusion.

[revised manuscript text omitted]

Commented [HJ11]: The numbers do match, but I will ensure the number is presented exactly as in the table Page 12, lines 4-20: Changed "65%" to "65.3"

Commented [HJ12]: Page 12, line 12: Changed "Table 1" to "Table 2"

Commented [HJ13]: It can due to the very irreversible nature of degradation Page 12, line 15: Changed "presumably" to "a possible indicator"

disturbance and rates of degradation (detected here by low LNS) has been noted by Hill et al. (2005) and Kairis et al. (2015) and specifically in the BDT by McKeon et al. (2009). Independent evidence for anthropogenesis presented here includes correlation with the VAST map which, although not a map of vegetation degradation, does
5    distinguish varying degrees of human-related modification of native vegetation (Thackway and Lesslie, 2005). The good agreement of ranks of average LNS and the VAST classes (Table 8) is evidence that LNS was able to separate human-related degradation from natural variation, at least up to the end of the period of time used for the VAST map (2011). In addition, there was qualitative evidence from visual
10   inspection of high resolution remotely-sensed imagery, such as abrupt differences across station boundaries (e.g. Figure 5b & Figure 5c) and coincidences of visible disturbance around livestock water points. The relationship between degradation, accelerated rates of erosion, and reduced vegetation cover is well-known (Lal, 2001) and erosion is the most widespread and recognizable characteristics of land degradation (Ravi et al., 2010), also a
15   primary impact on loss of soil carbon (Rajan et al., 2010). In the present study, there was a strong overall correlation of average LNS with hillslope erosion and gully density (Figure 7). In the BDT others have linked erosion with poor grazing management (Bartley et al., 2006) and unsustainable agricultural production (Montgomery, 2007).

        Assigning causal relationships to land degradation and natural or anthropogenic
20   factors is difficult due to the close coupling between humans and their environment (Reynolds et al., 2007). The LNS procedure offers one approach that attempts to isolate actual degradation of NPP from less favorable environmental conditions. However, without additional data on land usage, such as livestock numbers and management practices, the causes of the reductions by human-related activities are hard to determine
25   (Bastin et al., 2012). The most commonly-cited management practices to reduce degradation are reduction in domestic livestock, reduction of feral herbivores, removal of watering points (Bastin et al., 2012; Fensham and Fairfax, 2008; Silcock and Fensham, 2013), fallowing (Bastin et al., 2012; Bastin et al., 1993), or by encouraging vegetation that is particularly resistant to overgrazing or able to recover quickly after intense grazing
30   (Bastin et al., 2012; McKeon et al., 2004b; Smith et al., 2007). Additional data are needed to interpret low LNS, particularly with field observation.

        Given the extremely large areas of provincial, national, regional and global degradation that are frequently stated (Bai et al., 2008; Bridges and Oldeman, 1999; Kassas, 1995; Oldeman, 1994; UNEP, 1997; Zika and Erb, 2009) and the far-reaching
35   effects of degradation on human livelihoods (Adeel, 2008; UNCCD, 1994), rigorous, quantitative and objective measurements are urgently needed. While reduction of NPP is a single type of degradation, it is a quantitative measure of the outcome of most forms of degradation relevant to human needs - but not all (e.g. loss of palatable species with no change in NPPAsner and Heidebrecht, 2005). The widespread occurrence of

**Commented [HJ14]:** The evidence were the results from table 8.
Page 12, line 29: Changed "strong correlation" to "good agreement" to be more precise.

[revised manuscript text omitted]